# Hemispheric divergence of interoceptive processing across psychiatric disorders

**Emily M Adamic[1,2], Adam R Teed[1], Jason Avery[3], Feliberto de la Cruz[4], Sahib Khalsa[1,5]***

[1]Laureate Institute for Brain Research, Tulsa, United States; [2]Department of Biological Sciences, University of Tulsa, Tulsa, United States; [3]Laboratory of Brain and Cognition, National Institute of Mental Health, Bethesda, United States; [4]Laboratory for Autonomic Neuroscience, Imaging, and Cognition (LANIC), Department of Psychosomatic Medicine and Psychotherapy, Jena University Hospital, Jena, Germany; [5]Department of Psychiatry and Biobehavioral Sciences, Semel Institute for Neuroscience and Human Behavior, David Geffen School of Medicine, University of California at Los Angeles, Los Angeles, United States

## eLife assessment

This **fundamental** study provides **compelling** evidence for dysgranular insular involvement in top-down and bottom-up interoceptive processing by building on previous evidence using state-of-the-art methods. Its translational application in ADE patients corroborates the assumption that the mid-insula may indeed be a locus of 'interoceptive disruption' in psychiatric disorders, which underscores the study's high relevance for both body-brain as well as clinical research.

***For correspondence:**
skhalsa@mednet.ucla.edu

**Competing interest:** The authors declare that no competing interests exist.

**Abstract** Interactions between top-down attention and bottom-up visceral inputs are assumed to produce conscious perceptions of interoceptive states, and while each process has been independently associated with aberrant interoceptive symptomatology in psychiatric disorders, the neural substrates of this interface are unknown. We conducted a preregistered functional neuroimaging study of 46 individuals with anxiety, depression, and/or eating disorders (ADE) and 46 propensity-matched healthy comparisons (HC), comparing their neural activity across two interoceptive tasks differentially recruiting top-down or bottom-up processing within the same scan session. During an interoceptive attention task, top-down attention was voluntarily directed towards cardiorespiratory or visual signals. In contrast, during an interoceptive perturbation task, intravenous infusions of isoproterenol (a peripherally-acting beta-adrenergic receptor agonist) were administered in a double-blinded and placebo-controlled fashion to drive bottom-up cardiorespiratory sensations. Across both tasks, neural activation converged upon the insular cortex, localizing within the granular and ventral dysgranular subregions bilaterally. However, contrasting hemispheric differences emerged, with the ADE group exhibiting (relative to HCs) an asymmetric pattern of overlap in the left insula, with increased or decreased proportions of co-activated voxels within the left or right dysgranular insula, respectively. The ADE group also showed less agranular anterior insula activation during periods of bodily uncertainty (i.e. when anticipating possible isoproterenol-induced changes that never arrived). Finally, post-task changes in insula functional connectivity were associated with anxiety and depression severity. These findings confirm the dysgranular mid-insula as a key cortical interface where attention and prediction meet real-time bodily inputs, especially during heightened awareness of interoceptive states. Furthermore, the dysgranular mid-insula may indeed be a 'locus of disruption' for psychiatric disorders.

## Introduction

Maintaining the body in a physiologically optimal state is a primary function of the mammalian nervous system. The neurovisceral architecture accomplishing such feats depends on the hierarchical central processing of peripheral internal body signals and operates hierarchically and across heterogenous time scales (*Smith et al., 2017*), automatically returning the internal state to a setpoint (i.e. homeostasis) (*Cannon, 1939*) or shifting the internal state away from a setpoint in anticipation of future deviations (i.e. allostasis) (*Sterling, 2012*). Importantly, these regulatory operations may depend on the unconscious as well as conscious monitoring of interoceptive states. Interoceptive sensations may even dominate awareness during significant physiological disturbances, motivating behaviors that restore bodily states to homeostatic equilibrium (*Khalsa and Lapidus, 2016*) and forming the basis for emotional experience and numerous cognitive processes (*Adolfi et al., 2017*; *Carvalho and Damasio, 2021*; *Critchley and Garfinkel, 2017*; *Uddin et al., 2014*). In turn, dysfunctional interoception has been implicated in the pathophysiology of psychiatric (*Bonaz et al., 2021*; *Khalsa et al., 2018*) and neurological disorders (*Birba et al., 2022*; *Drane et al., 2020*), underscoring the importance of identifying the form and function of interoceptive neurocircuitry in humans, particularly in relation to a conscious experience of the inner body.

Conscious interoceptive perceptions must, to a considerable extent, reflect the organism's current physiological state. This is hierarchically conveyed to the brain by cascades of peripheral interoceptive inputs, from vagal and spinal afferent nerves relaying signals transduced from interoceptors located throughout the major internal organ systems (e.g. cardiovascular, respiratory, gastrointestinal, urogenital), as well as via direct chemo humoral signaling (*Berntson and Khalsa, 2021*). Ascending interoceptive pathways converge via brainstem and thalamic nuclei onto the insular cortex, a structurally and functionally diverse region suggested to be the integrative hub for interoceptive awareness, emotional feelings, and overall body awareness (*Craig, 2002*; *Craig, 2009*) (for discussions of additional pathways and cortical targets see *Berntson and Khalsa, 2021*; *Khalsa et al., 2009*; *Mayeli et al., 2023*). A principal role for the insular cortex in interoceptive awareness is supported by previous functional neuroimaging studies from our group, which have highlighted the insula as the primary region dynamically tracking pharmacological perturbations of the heart and lungs in both healthy individuals (*Hassanpour et al., 2016*; *Hassanpour et al., 2018*), and those with generalized anxiety disorder (*Teed et al., 2022*) (for fMRI meta-analyses of cardiac and gastrointestinal interoception findings see *Schulz, 2016* and *Halani et al., 2020*, respectively). Our studies have identified multiple subregions of the insula in this 'bottom-up' processing of afferently relayed cardiorespiratory signals, including the dysgranular mid and granular posterior sectors, in general agreement with existing theoretical (*Barrett and Simmons, 2015*; *Fermin et al., 2022*) and cytoarchitectonic (*Evrard, 2019*) arguments regarding their function.

While cortical sensitivity to ascending signals is presumptively required for conscious detection of internal physiological shifts in humans, accumulating evidence also suggests that subjective perceptions (in any sensory modality) are not simply veridical reflections of these ascending inputs, faithfully transmitted and reproduced in a dormant sensory cortex (*Clark, 2013*; *Corbetta et al., 2008*; *Mechelli et al., 2004*; *Pace-Schott et al., 2019*). Instead, intrinsic 'top-down' processes also appear to modulate the activity of sensory cortices (including the insula), potentially altering the resulting percept in the process. For example, internal models of the body that anticipate the sensory consequences of descending regulation can adjust the precision-weighting of prediction error signals in a context-specific manner, increasing or decreasing the sensory processing of interoceptive inputs (*Clark, 2016*; *Petzschner et al., 2021*; *Pezzulo et al., 2015*; *Seth and Friston, 2016*). Extending beyond these model-based influences are goal-directed activities (also described previously as the 'attentional spotlight' effect *Brefczynski and DeYoe, 1999*), whereby focusing voluntary attention towards certain environmental signals not only alters their conscious experience but selectively enhances neural activity in the responsive area of cortex. With respect to interoception, such goal-directed shifts in attention towards endogenous visceral signals have consistently identified activation of the mid insular cortex in healthy individuals (*Avery et al., 2017*; *Simmons et al., 2013*), and revealed abnormal mid insula activity associated with symptoms in those with eating disorders (*Kerr et al., 2016*), depression (*Avery et al., 2014*; *Burrows et al., 2022*), and substance use disorders (*Stewart et al., 2020*).

Conscious perceptions of interoceptive signals are thus underpinned by reciprocal interactions between bottom-up, ascending interoceptive afferents and the coupled, top-down descending efferent systems that enact adaptive regulation and attentional control (*Berntson and Khalsa, 2021*). Based on the available evidence, the insula is uniquely positioned to serve as a cortical interface between these two information streams for the processing of interoceptive signals. The highly interconnected nature of the mammalian insula with other cortical (prefrontal *Mufson and Mesulam, 1982*, somatosensory *Augustine, 1996*, cingulate *Mufson and Mesulam, 1982* cortices), subcortical (amygdala, thalamus *Gehrlach et al., 2020*), and brainstem (*Gogolla, 2017*) areas facilitates the ability for salient bottom-up activity to recruit conscious attention mechanisms and/or give rise to perceptions and feelings that can adaptively drive behavior and promote homeostatic regulation (*Evrard, 2019*). Furthermore, disrupted processing in this region may be characteristic of individuals with certain psychiatric disorders. In a previous human functional magnetic resonance imaging (fMRI) meta-analysis, abnormal activation of the left dorsal mid-insula differentiated healthy individuals from groups of individuals with anorexia nervosa, anxiety disorders, depression, bipolar disorder, or schizophrenia, across tasks involving the bottom-up modulation of pain, dyspnea, or hunger or the top-down manipulation of interoceptive attention (*Nord et al., 2021*). This insular 'locus of disruption' was inferred as a possible transdiagnostic and domain-general neuro marker of psychopathology, and the finding advanced the field by pointing toward the mid-insula as an important area for interoceptive dysfunction and a putative target for clinical intervention (*Paulus and Khalsa, 2021*). Despite the theoretical and empirical convergence related to bottom-up and top-down processing in the insula, no study to date has directly investigated both interoceptive processing streams together in the same sample.

Here, we conducted a pre-registered conjunction analysis of two established neuroimaging tasks preferentially manipulating the assessment of top-down and bottom-up interoception (*Adamic et al., 2021*). Our hypotheses, based on a voxelwise whole-brain approach, were threefold: (1) broad regions of the dysgranular mid insula would be co-activated when (a) engaging top-down interoceptive attention under physiological resting conditions and (b) during bottom-up perturbation of peripheral interoceptive signals via the adrenergic drug isoproterenol, (2) the agranular anterior insula would

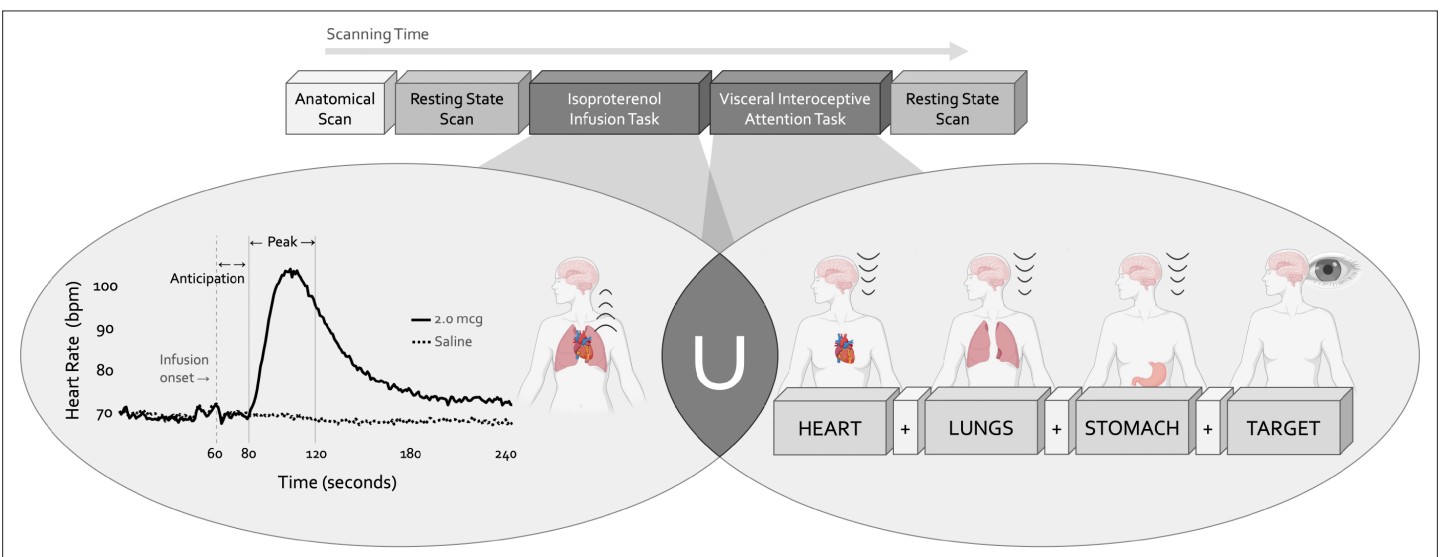

**Figure 1.** Experimental design. During a single functional neuroimaging scanning session two neuroimaging tasks were used to localize the convergence between bottom-up and top-down interoceptive processing. The isoproterenol infusion (ISO) task manipulated interoceptive input in a bottom-up manner, while the visceral interoceptive attention (VIA) task manipulated interoceptive attention in a top-down manner. During the ISO task, participants were asked to attend to their cardiorespiratory sensations while receiving double-blinded infusions of either isoproterenol 2.0 micrograms (mcg, solid line) or 0.5 mcg (not pictured), a rapidly acting peripheral beta-adrenergic agonist eliciting transiently increased cardiovascular and respiratory signals in a manner akin to adrenaline, or infusions of saline (dashed line), resulting in no physiological change. During the VIA task, on-screen cues directed participants to shift their attention towards naturally occurring body sensations from a particular internal organ (heart, lungs, or stomach, the interoceptive attention conditions) or the word 'TARGET' (the exteroceptive attention condition) that would flash at different intensities. No infusions were given during this task, so the body remained at physiological rest. Both tasks were preceded and followed by a resting state scan.

be preferentially selective to manipulations of interoceptive uncertainty, such as those occurring during saline infusions, and (3) relative to healthy individuals, interoceptive processing in the insula would show lateralized differences across a transdiagnostic group of individuals with anxiety, depression, and/or eating disorders. To extend the functional interpretation of convergence, the convergent dysgranular insula regions identified in the initial analysis were used as seeds in an exploratory whole-brain resting state functional connectivity (FC) analysis. Finally, divergent voxels selective to the engagement of top-down expectancies about the body were isolated, focusing on temporal windows of the ISO task when the anticipation of potential interoceptive change was maximal. We predicted that the agranular insula would be preferentially engaged during these windows, and not engaged during the peak of perturbation or during interoceptive attention alone.

## Results

Of the 127 total participants included in this study, 57 ADE individuals and 46 HC individuals fulfilled the criteria for high-quality brain imaging data with both tasks. In line with the predominant female prevalence of these disorders, most of the participants were female. To obtain equal-sized groups for group-level comparison, propensity matching on age, self-reported sex, and body mass index resulted in 46 individuals per group (*Figure 1*). After propensity matching, the two groups were not significantly different regarding age, body mass index, and sex, but they differed significantly on trait measures of anxiety (State-Trait Anxiety Inventory; STAI), anxiety sensitivity (Anxiety Sensitivity Index; ASI, Total and subscales), depression (Patient Health Questionnaire-9; PHQ-9) (*Kroenke et al., 2001*), and eating disorders (Sick, Control, One, Fat, Food questionnaire; SCOFF) (*Kutz et al., 2020*). All participants in the ADE group exhibited some form of psychiatric comorbidity. Across the group, 67% had a diagnosis of GAD (current or lifetime), 72% of major depression (current or remitted), and 35% of anorexia nervosa (current or lifetime). Most ADE participants reported taking psychiatric medication (see *Table 1* for details).

### Convergence analysis

Bilateral patches of the ventral mid-insular cortex were commonly activated during the ISO and VIA tasks (*Figure 2*) (*Table 2*). Quantification of the spatial extent of this convergence in the pre-defined probabilistic agranular, dysgranular, and granular subregions revealed that these convergent clusters were localized primarily to the dysgranular insula bilaterally (*Figure 3A*), but with a hemispheric asymmetry across groups: the ADE group showed a greater proportion of overlapping voxels within the left dysgranular insula ($\chi^2_1$ = 26.7, p<0.001 for group comparison) whereas the HC group showed a greater number of overlapping voxels within the right dysgranular insula ($\chi^2_1$ = 26.7, p<0.001 for group comparison) (*Figure 3B*). Correspondingly, the spatial similarity of these convergently activated voxels between the groups was lower for the left dysgranular insula (Dice coefficient of 0.58) than for the right (Dice coefficient of 0.78) (*Figure 2*). In addition to these group hemispheric differences in spatial extent, there was a lower amount of overlap between the two tasks in the left dysgranular insula for both groups (left hemisphere overlap coefficients: 0.42 and 0.63 for HC and ADE, respectively; right hemisphere overlap coefficients: 0.83 and 0.78, for HC and ADE, respectively).

Despite these group differences in the spatial extent and spatial similarity of convergence, the magnitude of activation within convergent dysgranular subregions during the Peak of perturbation (ISO 2.0 mcg dose) or Heart and Lung attention (VIA task) was not different between groups ($F_{1,90}$=0.64 and 0.05, p=0.43 and 0.82 for main effect of group in ISO and VIA tasks, respectively). However, during both of these task conditions, activation was greater within the right hemisphere than the left ($F_{1,90}$=6.13 and 5.10, p=0.015 and 0.027 for main effect of hemisphere during ISO and VIA tasks, respectively; $t_{90}$=–2.48 and –2.25, p=0.02 and 0.03 for post-hoc contrasts between the left versus right hemisphere when collapsed across groups) (*Figure 4A*). This hemispheric effect on activation magnitude was specific to interoceptive processing, as no similar hemispheric difference was seen during the exteroceptive condition of the VIA task ($F_{1,90}$=0.68, p=0.41). Furthermore, the magnitude of activation within the right (but not the left) dysgranular convergent region during the peak of cardiorespiratory perturbation was associated with the intensity of real-time cardiorespiratory sensations (i.e. mean dial rating) during this period when examined across the entire sample (Pearson's r=0.12 and 0.21, p=0.25 and 0.046 for the left and right hemispheres, respectively; *Figure 4B*).

**Table 1.** Demographic information for the transdiagnostic anxiety, depression, and/or eating disorders (ADE) group and the propensity-matched healthy comparison (HC) group. GAD=generalized anxiety disorder, MDD=major depressive disorder, AN=anorexia nervosa, PTSD=post-traumatic stress disorder, OCD=obsessive-compulsive disorder, SUD=substance use disorder, BMI=body mass index, STAI=State-Trait Anxiety Inventory, ASI=Anxiety Sensitivity Index, with a Total Score, and Physical, Cognitive, and Social subscales, PHQ-9=depression module of the Patient Health Questionnaire, SCOFF='Sick, Control, One, Fat, Food' eating disorder questionnaire.

| | HC (n=46) | ADE (n=46) | t | df | p-value |
|---|---|---|---|---|---|
| Sex | 41 Female | 45 Female | - | - | - |
| | 5 Male | 1 Male | | | |
| Age | 24 (5) | 25 (7) | 0.57 | 80.49 | 0.57 |
| BMI | 23.48 (3.26) | 23.34 (3.33) | –0.20 | 89.97 | 0.84 |
| STAI | 30.0 (1.1) | 56.0 (1.4) | –14.43 | 83.01 | **<0.0001** |
| ASI-Total | 7.39 (5.26) | 13.98 (8.89) | 8.89 | 57.47 | **<0.0001** |
| ASI-Physical | 1.15 (1.59) | 5.19 (5.81) | 5.81 | 53.38 | **<0.0001** |
| ASI-Cognitive | 0.98 (1.47) | 7.41 (6.53) | 6.52 | 49.54 | **<0.0001** |
| ASI-Social | 5.26 (4.04) | 13.83 (6.08) | 7.96 | 78.28 | **<0.0001** |
| PHQ-9 | 0.78 (1.11) | 10.39 (5.98) | 10.72 | 48.12 | **<0.0001** |
| SCOFF | 0 (0) | 1.09 (1.23) | 6.01 | 45 | **<0.0001** |
| Diagnoses | No psychiatric disorders | 67% GAD | - | - | - |
| | | 72% MDD (current or remitted) | | | |
| | | 35% AN | | | |
| | | 13% Social anxiety disorder | | | |
| | | 11% PTSD | | | |
| | | 4% OCD | | | |
| | | 4% SUD | | | |
| Medication use | None | *Taking one or more psychiatric medication:* | - | - | - |
| | | Serotonergic[*]: 35% | | | |
| | | Histaminergic[†]: 11% | | | |
| | | Dopaminergic[‡]: 7% | | | |
| | | SNRI[§]: 7% | | | |
| | | Benzodiazepene[¶]: 4% | | | |
| | | Miscellaneous[**]: 2% | | | |
| | | Beta-blocker[††]: 2% | | | |

[*]Escitalopram, Fluoxetine, Trazodone, Sertraline, Paroxetine, Duloxetine.
[†]Lamotrigine, Hydroxyzine, Doxepin.
[‡]Olanzapine, Aripiprazole, Quetiapine.
[§]Atomoxetine, Buproprion, Desvenlafaxine.
[¶]Clonazepam, Alprazolam.
[**]Vortioxetine.
[††]Propranolol (as needed usage).

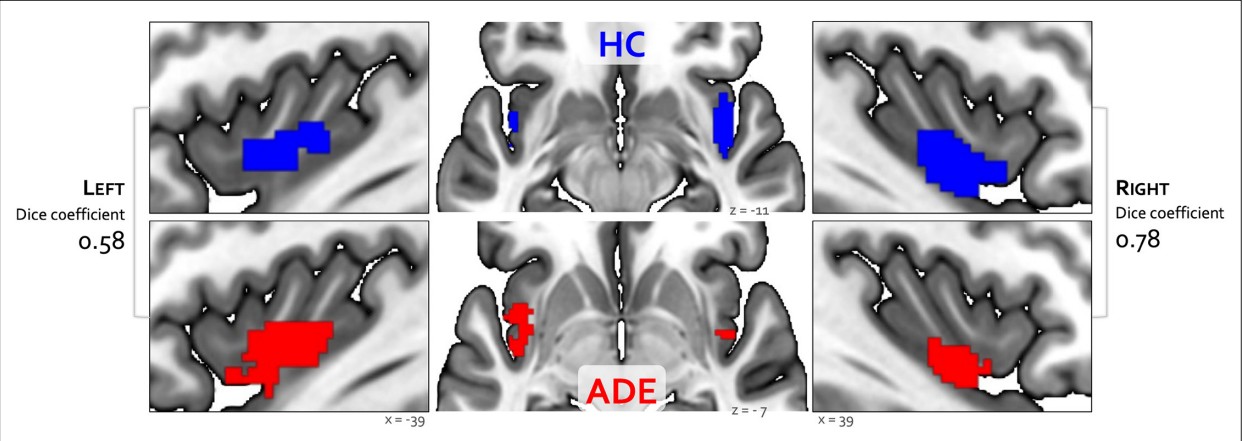

**Figure 2.** Convergence between bottom-up and top-down interoceptive processing in healthy comparisons (HC, top row) and individuals with anxiety, depression, and/or eating disorders (ADE, bottom row). Each convergence map reflects the only cluster of voxels that were co-activated across the whole brain during the isoproterenol nfusion infusion (ISO) task, involving perturbation of cardiorespiratory signals, and the visceral interoceptive attention (VIA) task, involving goal-directed interoceptive attention towards cardiac and respiratory signals at rest. Numbers on the left and right reflect the Dice similarity coefficient, which is used to quantify the degree of spatial overlap between groups. Relative to HCs, the ADE group showed lower spatial similarity for the left insular cortex (Dice coefficient of 0.58), than the right insular cortex (Dice coefficient of 0.78).

Conversely, PSC in either convergent hemisphere during perturbation was not associated with objective measures of physiological arousal (i.e. mean change in heart rate, Pearson's $r$=0.11 and 0.09, p=0.31 and 0.40 for left and right hemispheres, respectively), post-scan ratings of cardiac intensity (Pearson's $r$=0.16 and 0.20, p=0.15 and 0.06), post-scan ratings of respiratory intensity (Pearson's $r$=0.12 and 0.06, p=0.26 and 0.58 for left and right hemispheres, respectively), post-scan ratings of state anxiety (Pearson's $r$=0.14 and 0.08, p=0.20 and 0.49 for left and right hemispheres, respectively), or trait measures of anxiety sensitivity (i.e. ASI-Total, Pearson's $r$=0.12 and 0.16, p=0.25 and 0.13 for left and right hemispheres, respectively).

Finally, pre- to post-task resting state functional connectivity of the convergent regions showed a pattern of differential change in the ADE compared to HC group: functional connectivity between the right dysgranular convergent cluster and left middle frontal gyrus increased in the ADE group only, while the HC group saw no differences (p<0.05, corrected; *Figure 5A and B*). This was the only significant cluster showing connectivity differences between groups. When looking at this insular-IFG connection across the entire sample, the magnitude of functional connectivity change between these two areas was associated with trait anxiety (Pearson's $r$=0.43 and 0.36 for OASIS and GAD7, respectively, p<0.001) and trait depression (Pearson's $r$=0.38, p<0.001 for PHQ-9) (*Figure 5C*) but not the magnitude of ISO-induced heart rate (Pearson's $r$=–0.05, p=0.66) or dial rating (Pearson's $r$=0.01, p=0.90) at the 2.0 µg dose.

## Divergence analysis

The Anticipation period of the ISO task engaged large swaths of the bilateral agranular insular cortex that were not activated at any point during the VIA task (*Figure 6A*) (*Table 3*). During this period, the

**Table 2.** Insular regions demonstrate convergent processing between top-down and bottom-up interoception.

| | MNI coordinates (center of mass) | | | Volume (mm³) |
|---|---|---|---|---|
| | X | Y | Z | |
| Left Insula: ADE | –39 | –0.1 | –8.2 | 1736 (217 voxels) |
| Left Insula: HC | –40 | –1.3 | –4.8 | 1240 (155 voxels) |
| Right Insula: ADE | 39 | 2.3 | –12.9 | 1088 (136 voxels) |
| Right Insula: HC | 39.9 | 2.2 | –10.3 | 1960 (245 voxels) |

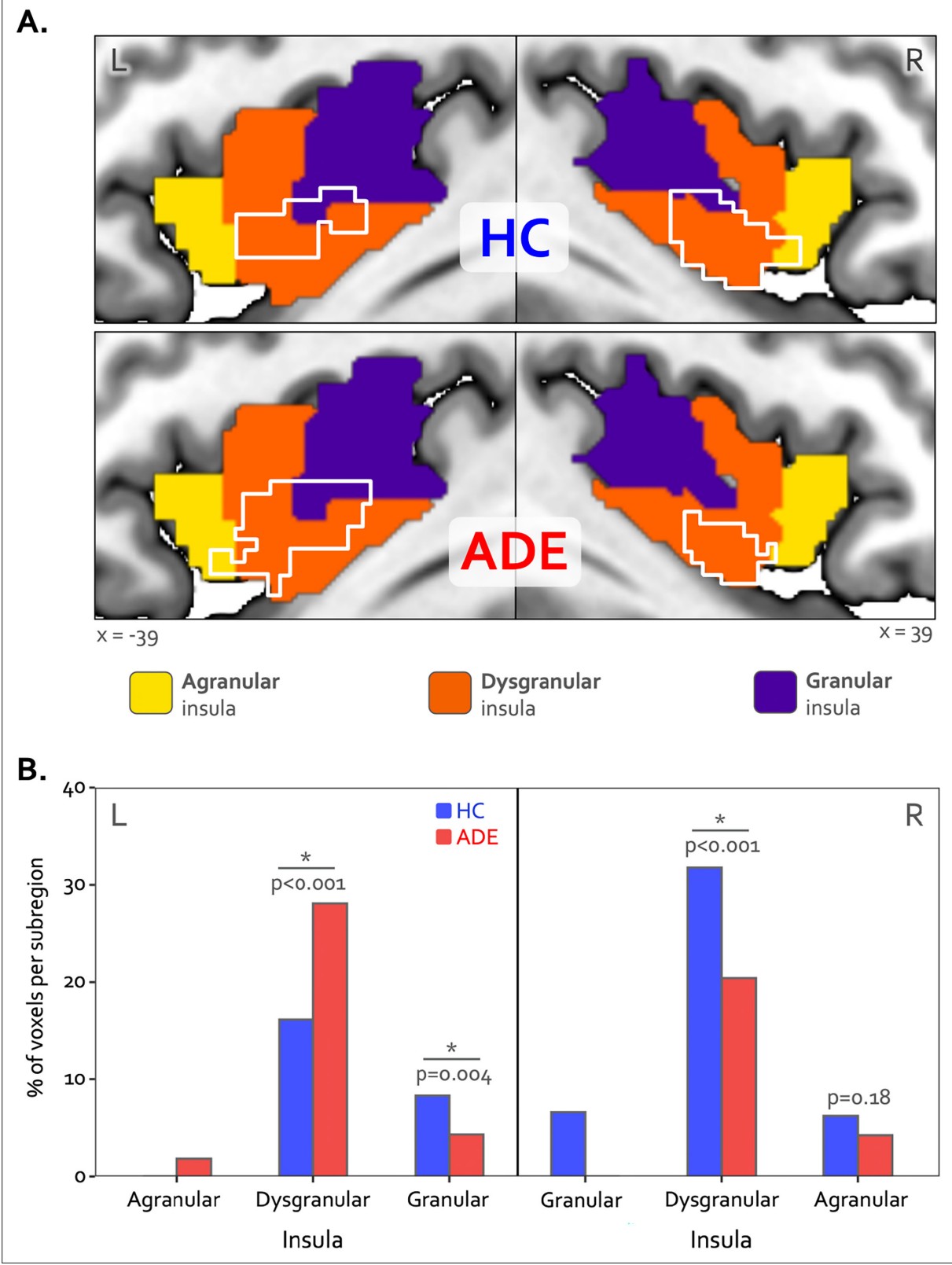

**Figure 3.** Hemispheric divergence of bottom-up and top-down interoceptive processing across psychiatric disorders. (**A**) Hemisphere-specific convergence maps (white outlines) overlaid on a tripartite probabilistic cytoarchitectonic division of the insular cortex. (**B**) Quantification of convergence within each cytoarchitectonic subregion (i.e. number of co-activated voxels in relation to the total number of voxels within that subregion), showing asymmetric proportional voxel co-activation in the left and right dysgranular insula between the groups. *indicates significant group difference for the proportion of co-activated voxels in each subregion via chi-square test. HC: healthy comparison. ADE: anxiety, depression, and/or eating disorders.

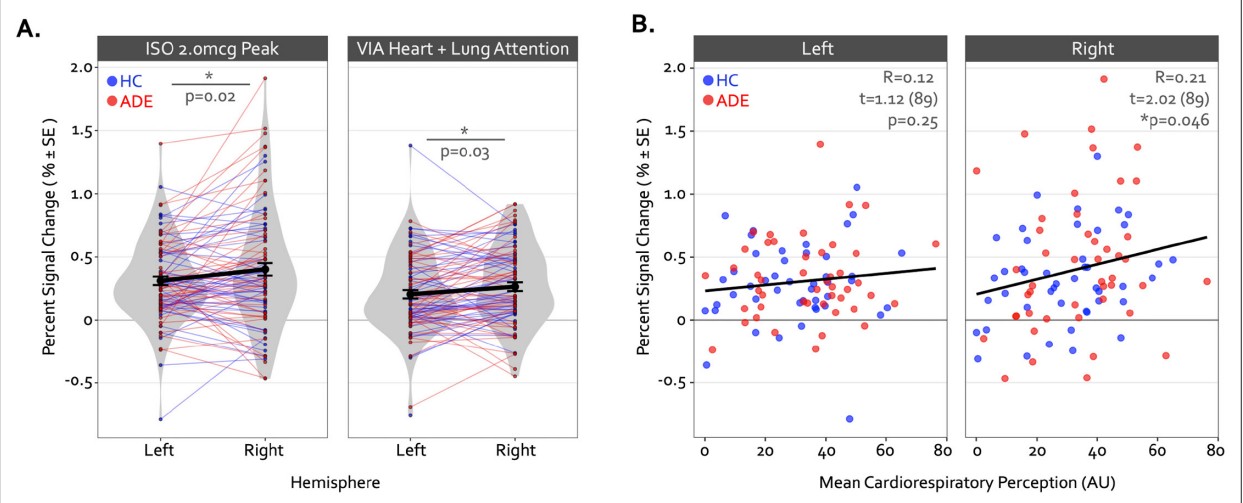

**Figure 4.** Hemispheric differences in the activation magnitude of the convergent dysgranular insula subregion. (**A**) Co-activated voxels within the right dysgranular insula subregion exhibited a greater degree of percent signal change during the Peak period of the 2.0 mcg dose in the isoproterenol infusion (ISO) task (left sub-panel) and during the Heart and Lung attention condition in the visceral interoceptive attention (VIA) task (right sub-panel). As there were no main effects of group, the mean percent signal change is collapsed across groups. *indicates p<0.05 for the left versus right post-hoc contrast, following a main effect of the hemisphere in the linear mixed-effects regression. (**B**) Across both groups, the mean percent signal change within the right but not left convergent dysgranular insula subregion was correlated with real-time cardiorespiratory intensity ratings during the Peak period of the 2.0 mcg dose in the ISO task. *indicates p<0.05 for the Pearson's R correlation coefficient, separately for each hemisphere.

right hemisphere was activated to a greater spatial extent than the left in both groups ($\chi^2_1$ = 22.7 and 7.3, p<0.001 and p=0.007 for left vs. right in the HC and ADE groups, respectively), and to a greater spatial extent in HC versus ADE individuals in the right hemisphere only ($\chi^2_1$ = 7.3, p=0.007 for group comparison). There was no group difference in the spatial extent in the left hemisphere ($\chi^2_1$ = 2.0, p=0.15) (*Figure 6B*). Notably, these spatial patterns of activation were similar between groups (Dice coefficient = 0.86 in the left and 0.87 in the right).

During the Peak period of the Saline dose, the spatial extent of activation in the agranular insula increased to include significantly more voxels, for both groups and hemispheres ($\chi^2_1$ = 75.3 and 71.4, p<0.001 for Anticipation vs. Peak in the left and right hemispheres in HC; $\chi^2_1$ = 36.3 and 12.1, p<0.001 for Anticipation vs. Peak in the left and right hemispheres in ADE) (*Figure 6B*). This activation expansion occurred to a greater degree in the HC versus ADE group ($\chi^2_1$ = 17.2 and 70.3, corrected p<0.001 for HC versus ADE in the left and right hemispheres, respectively). Finally, agranular insula recruitment within the HC group during the Saline Peak period was shifted towards the right hemisphere (i.e. more active voxels in the right than the left, $\chi^2_1$ = 24.3, p<0.001), while the ADE group showed no hemispheric difference during this period ($\chi^2_1$ = 0.10, p=0.75). Despite this difference in number of activated voxels, the spatial patterns of activation were still largely overlapping between groups (Dice coefficient = 0.88 in the left, 0.82 in the right).

Along with expansions in the spatial extent of activation, the signal intensity in the agranular insula increased from the Anticipation period to the Saline Peak period ($F_{1,270}$=28, p<0.0001 for the main effect of epoch), with no differences between groups ($F_{1,270}$ = 0.25, p=0.62 for the main effects of group) (*Figure 6C*). This increase occurred in both hemispheres ($t_{270}$=2.7 and 4.8, p=0.008 and <0.001 for Saline Peak – Anticipation contrast in the left and right hemispheres, respectively, when collapsed across groups). Notably, the right hemisphere was activated more strongly than the left during the Saline Peak period ($t_{270}$=–3.0, p=0.003), whereas there was no difference across hemispheres in the Anticipation window ($t_{270}$=–0.8, p=0.41). The magnitude of agranular insula activation was not related to trait measures of anxiety sensitivity in either epoch or hemisphere when examined transdiagnostically across groups (ASI * hemispheric interaction effect: $F_{1,90}$ = 1.9, p=0.17 for Anticipation, and $F_{1,90}$ = 1.7, p=0.19 for Saline Peak).

## Physiological response and perceptual ratings

During the 2.0 mcg dose there were large perturbations in heart rate in both groups, and while the ADE appeared to have a slightly greater response on average, this reached statistical significance via

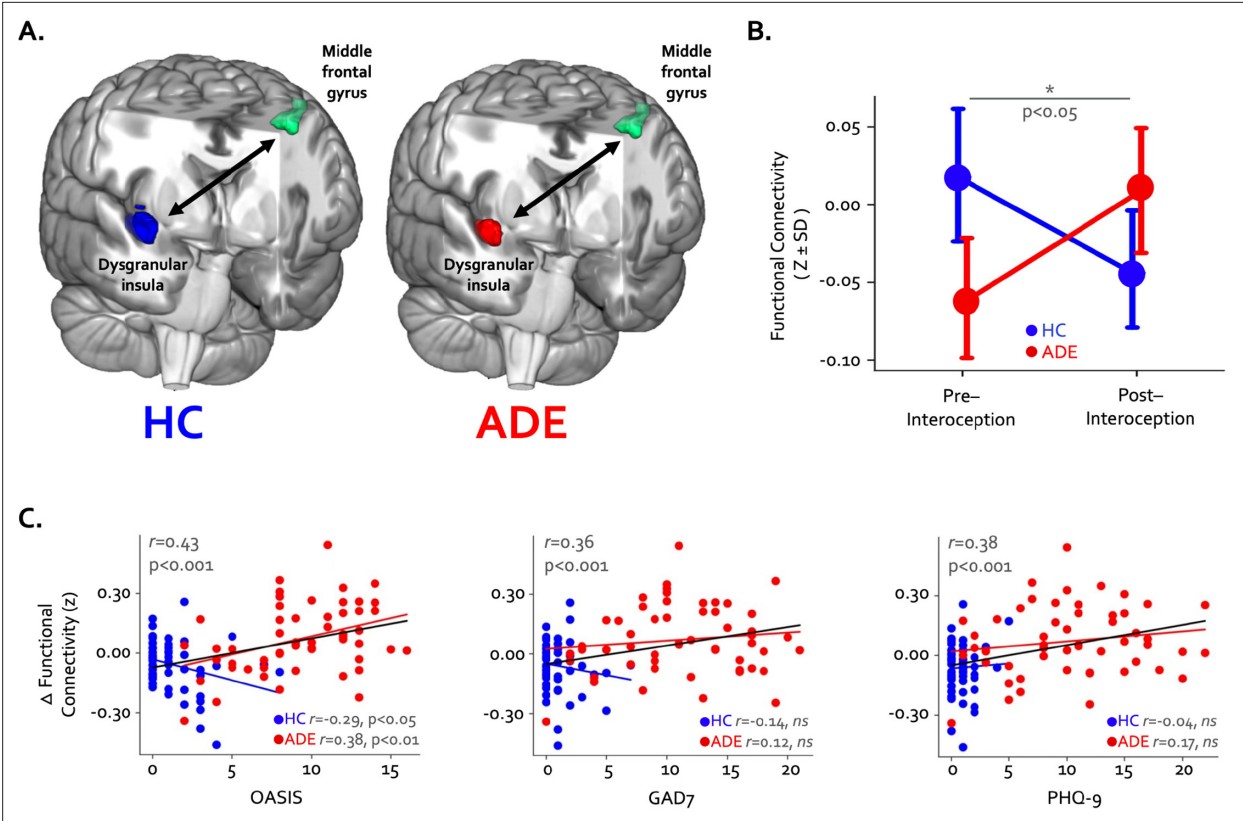

**Figure 5.** Changes in whole-brain functional connectivity of convergent dysgranular insula subregions following the performance of the interoceptive tasks. (**A**) Functional connectivity between the right convergent dysgranular insula subregion and the left middle frontal gyrus showed opposing effects in the anxiety, depression, and/or eating disorders (ADE) versus healthy comparisons (HC) groups. (**B**) While functional connectivity between these regions increased from baseline following interoception in the ADE group, it decreased in the HC group, resulting in a significant interaction effect. (**C**) Across the entire sample changes in functional connectivity were associated with trait measures of anxiety and depression. When examined in each group individually, associations between connectivity change and trait anxiety occurred in opposite directions (left and middle panel).

permutation testing only during two continuous time points in the Peak period (98 s through 99 s, t=1.7 and 1.5, p=0.044 and 0.045, *Figure 7A*). However, the ADE group reported significantly higher real-time dial ratings for the latter portion of the Peak period extending into much of the ensuing recovery period (110 s through 142 s, t=1.8–2.3, p=0.008–0.042, *Figure 7B*). During the Saline dose of the ISO task, both heart rate and perceptual dial ratings remained near baseline levels. While heart rate did not differ between groups (t=–0.53–1.64, p=0.05–1.00), the HC group showed small but statistically significant increases in dial ratings compared to those of the ADE group for portions of the Peak period and the ensuing recovery period (t=–1.67 to –2.60, p=0.005–0.048) (*Figure 7A and B*). Although there were ISO-induced differences in continuous measures of cardiorespiratory sensation, there were no group differences in retrospective cardiac or respiratory intensity ratings for the 2.0 mcg infusion ($t_{80}$=1.5 and 1.2 for heart and lungs respectively, p=0.13 and 0.25) or the saline infusion ($t_{80}$=0.3 and 1.2, p=0.75 and 0.22) (*Figure 7C*). The ADE group reported greater retrospective anxiety than the HC group following infusions ($t_{85}$=4.8 and 3.8 for 2.0 mcg and Saline, respectively, p<0.001), with no differences in excitement ($t_{85}$=0.2 and 0.9, p=0.80 and 0.37 for 2.0 mcg and Saline, respectively). During the VIA task, the ADE group reported higher cardiac intensity ratings than the HC group during heart-focused attention ($t_{89}$=2.0, p=0.04) but lower exteroceptive intensity ratings during the target condition ($t_{90}$=2.7, p<0.01). There were no group differences in the intensity of reported lung or stomach sensations ($t_{89}$=1.6 and 1.4, p=0.10 and 0.16) (*Figure 7D*).

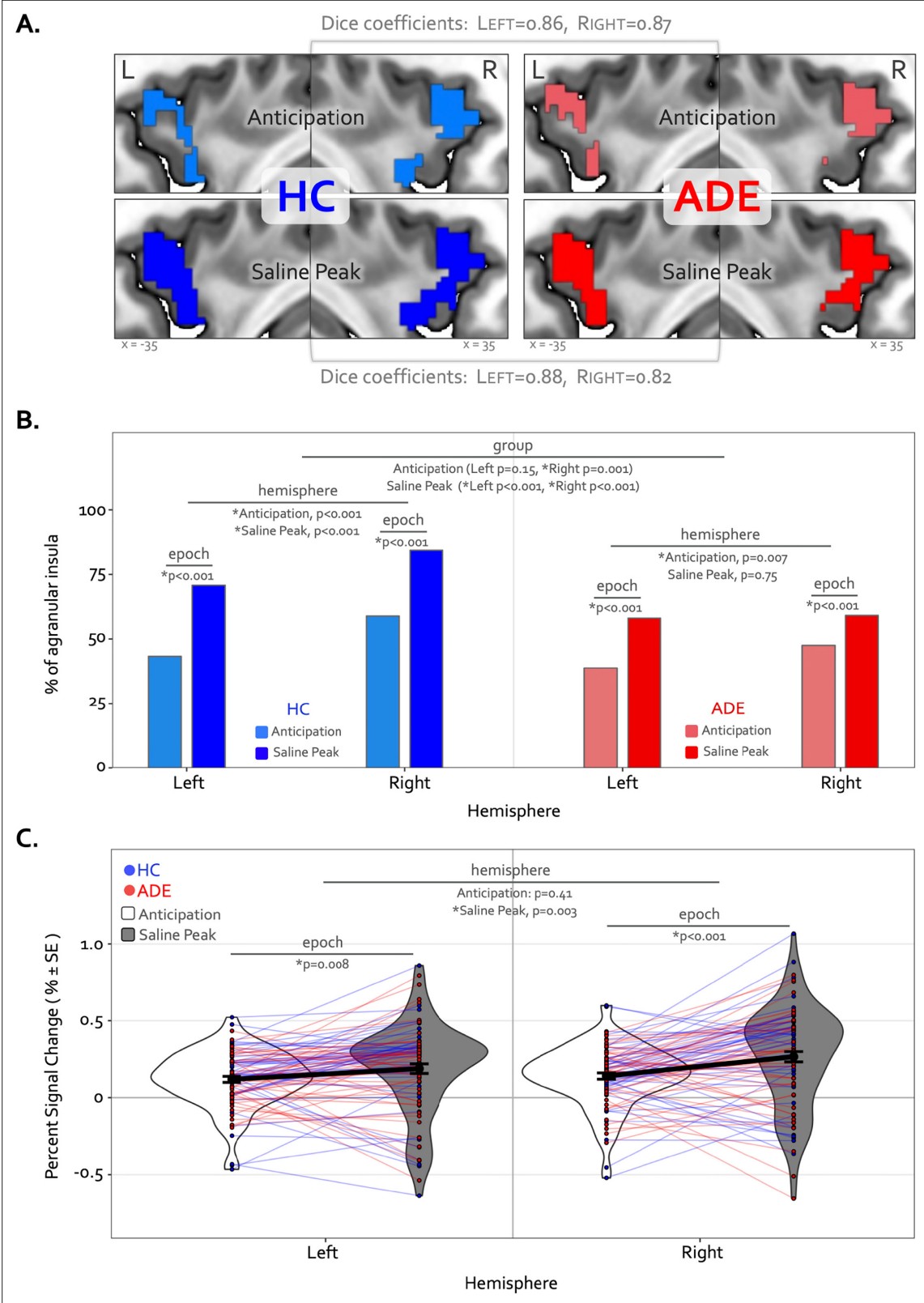

**Figure 6.** Divergence of top-down processing during interoceptive uncertainty in individuals with anxiety, depression, and/or eating disorders (ADE) and healthy comparisons (HC). (**A**) Selective activation of the bilateral agranular anterior insula during the Anticipation period (top) and Saline Peak period (bottom) of the isoproterenol (ISO) task. Both epochs demarcate periods of maximal expectancy about future changes in body state while the body remains at physiological rest (i.e. no ensuring evidence of heart rate or breathing rate increases). These brain areas showed activity only during

*Figure 6 continued on next page*

*Figure 6 continued*

these periods of the ISO task and were not active during the Heart and Lung attention part of the visceral interoceptive attention (VIA) task, indicating they have specialized roles. (**B**) Quantifying the spatial extent of this agranular activation (i.e. number of active voxels in relation to total number of voxels in that subregion) revealed that greater right hemisphere activation during the anticipation window occurred for both groups, and more so in the HC than the ADE group. During the peak period of Saline, this activation covered more of the bilateral agranular insula in both groups. However, the right hemisphere was activated more than the left in both groups, and the HC group exhibited more active voxels in the right hemisphere than the left, and more than the ADE group in both hemispheres, while the ADE group showed no hemispheric difference during this window. (**C**) Increased ISO-specific activation of agranular insular during the anticipation period versus the saline peak period. This pattern, occurring for both hemispheres and across both groups, was greater in the right than the left hemisphere.

## Discussion

The current study revealed overlapping patterns of neural activation within the dysgranular mid-insula during top-down and bottom-up interoceptive processing, corroborating previous studies examining each task independently (*Hassanpour et al., 2018*; *Simmons et al., 2013*), and supporting the notion of the insula as a key functional interface between goal-directed and sensory-driven interoceptive streams. Importantly, hemispheric differences in the patterns of dysgranular activation distinguished individuals with anxiety, depression, and eating disorders from healthy comparisons, as hypothesized. These hemispheric asymmetries, and the disparate spatial patterns within the left dysgranular insula, support the conceptualization of this subregion as a 'locus of disruption' for interoceptive symptomatology in these disorders (*Nord et al., 2021*). The hemispheric imbalance was concurrent with a heightened perceptual response in the clinical sample, as well as with changes in right dysgranular insula-to-left middle frontal gyrus functional connectivity following the interoceptive tasks. Finally, activity within the agranular insula was selective to periods of increased interoceptive expectancies, aligning with a prediction-generating role for this region and one that was distinct from the convergent effects seen in the dysgranular mid-insular subregion.

Top-down processing commonly refers to cognitive influences on perception, such as attention, memory, and expectation, that putatively help the brain to make sense of the stimuli it encounters. In contrast, bottom-up processing commonly refers to the stimulus-driven signals that are processed hierarchically as they reach the brain. With respect to interoception, top-down processing can refer to the brain's ability to predict or interpret how the internal body should feel (i.e. the expected interoceptive signals) based on the previous or current environmental context. In turn, bottom-up processing can refer to the actual state of the body in the current moment. For instance, top-down influences may include the anticipation of an upcoming meal leading to feelings of hunger, while bottom-up influences may refer to actual hunger signals from the body at lunch time, such as hormonal concentrations of ghrelin/leptin, and stomach contractions. The continuous balance between these processes determines the qualia of conscious perceptions and has been suggested to be key to health and the

**Table 3.** Agranular anterior insular regions demonstrate divergent processing during periods of increased bodily expectancies.

| | | MNI coordinates (center of mass) | | | Volume (mm³) |
|---|---|---|---|---|---|
| | | X | Y | Z | |
| Anticipation | Left: ADE | −31.6 | 17.9 | 4.8 | 1168 (146 voxels) |
| | | −31.9 | 9.2 | | 344 (43 voxels) |
| | Left: HC | −31.5 | 15.2 | −1.2 | 1688 (211 voxels) |
| | Right: ADE | 35.4 | 19.1 | 1.4 | 1648 (206 voxels) |
| | Right: HC | 37 | 13 | −9 | 1744 (218 voxels) |
| | | 35 | 7 | −19 | 368 (46 voxels) |
| Saline (Peak) | Left: ADE | −33.1 | 16.7 | −1.6 | 2264 (283 voxels) |
| | Left: HC | −33.2 | 15.7 | −2.7 | 2760 (345 voxels) |
| | Right: ADE | 35.4 | 18.6 | −0.9 | 2120 (265 voxels) |
| | Right: HC | 34.3 | 16.7 | −4.7 | 3024 (378 voxels) |

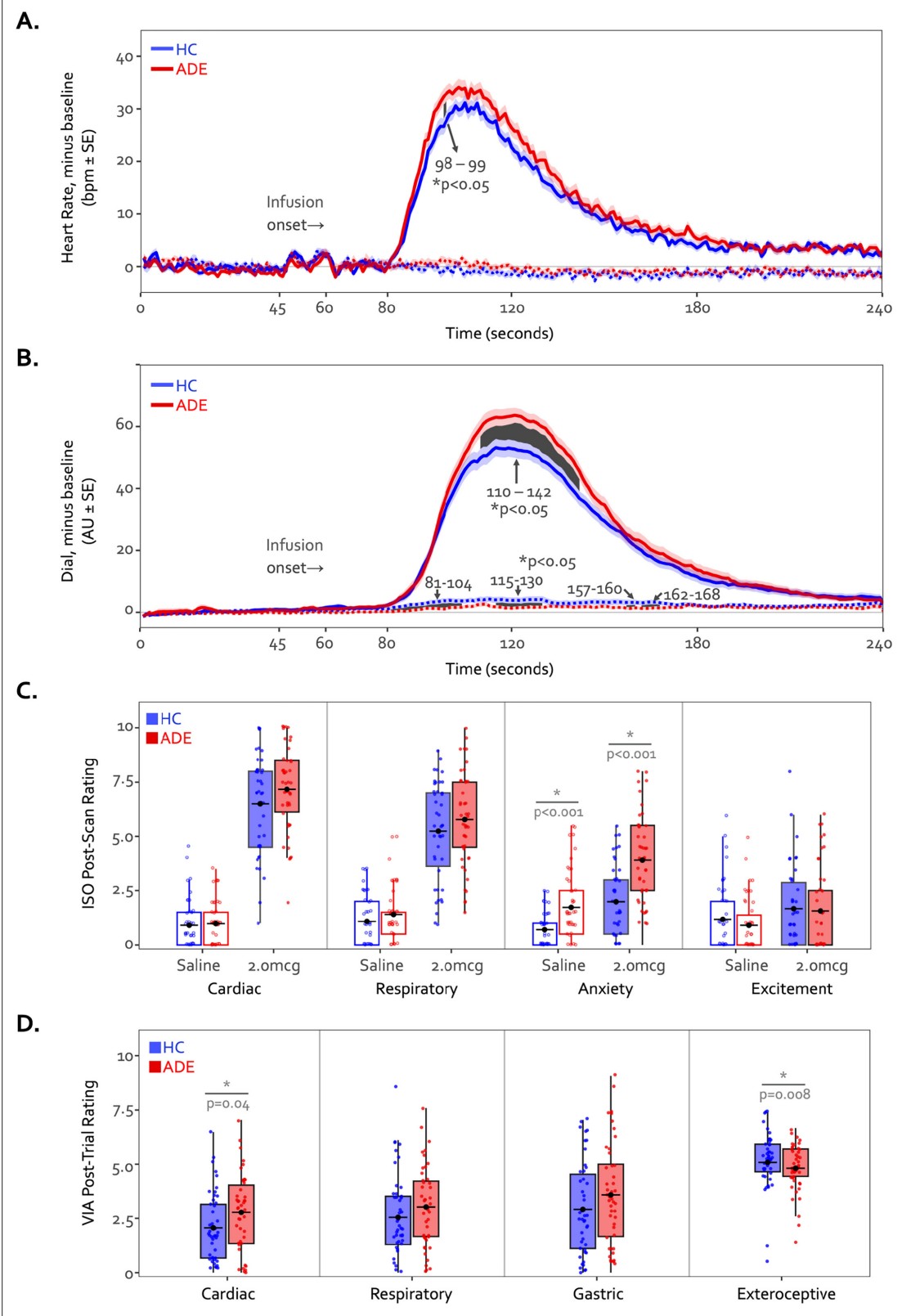

**Figure 7.** Time courses of heart rate (**A**) and continuous dial ratings (**B**) during the 2.0 mcg (solid lines) and Saline (dotted lines) doses of the isoproterenol (ISO) task. Heart rate changes observed during both infusions were generally similar between both groups across the majority of the infusion period. The cluster-based permutation analysis showed a 2 s instance of significantly higher heart rate in the anxiety, depression, and/or eating disorders (ADE) group starting at the 98th second of the 2.0 mcg dose only during the Peak period, followed by a prolonged increase in dial rating for

*Figure 7 continued on next page*

*Figure 7 continued*

large portions of the Peak and Recovery periods (110 s through 142 s, shown via the shaded area). No heart rate differences were observed during the Saline infusion. *indicates p<0.05 for significant group differences using permutation analysis. (**C**) Retrospective ratings on the ISO task showed no significant group differences in cardiac or respiratory intensity or excitement following saline or the 2.0 mcg infusion, while the ADE group reported increased anxiety following both infusions. (**D**) Sensation ratings during the visceral interoceptive attention (VIA) task. The ADE group reported higher cardiac intensity ratings than the healthy comparison (HC) group but lower exteroceptive ratings, with no group difference in respiratory or gastric sensation ratings. *indicates p<0.05 for a two-sample t-test between groups, separately for each condition. Where appropriate, both the p-value and Cohen's d are shown.

optimization of internal and subjective states (*Sterling, 2012*). However, as processes, they are rarely studied concomitantly in the manner conducted in this study. The convergence identified between top-down and bottom-up interoceptive processing within the dysgranular mid-insula suggests that this cortical subregion is an area where further detail regarding the reciprocal interactions underpinning interoceptive perceptions should be investigated, particularly in relation to psychiatric disorders involving disturbances of mood, anxiety, or appetite.

Although the current findings emphasize the role of interoceptive processing within dysgranular subregions of the insula, a number of experimental and analytic next steps are worth considering. Due to the broad systems-level of inference allowed by fMRI methodology and the relatively rudimentary block averaging analytic approach employed, additional measurements are warranted to establish the cellular and molecular mechanisms underlying the observed dysgranular co-activation. Such approaches would be important for verifying a true intersection of interoceptive neurocircuitry in which connecting top-down attention and prediction circuits directly modulate, or are modulated by, incoming sensory afferences. An alternative outcome could be identifying an overpassing flow of signals across these circuits within the same region. Implementing high-resolution approaches to imaging these processes via high-field fMRI on a trial-by-trial basis could inform the laminar architecture underpinning attention-driven modulation of incoming viscerosensory signals as well as allowing for computational modelling of these processes at the level of cortical laminar subfields (*Kasper et al., 2022*). In an interoceptive application of this analyses employing a respiratory bottom-up perturbation approach, activation of the anterior insula was associated with the prediction certainty but not prediction error for different inspiratory breathing loads, with parameters distinguishing between high versus low trait anxiety individuals (*Harrison et al., 2021*). This computational approach could be adapted for cardiorespiratory processing within the dysgranular insula. While typically restricted to high field strengths, such as 7 Tesla, there are emerging reports that laminar imaging can be performed at lower field strengths (*Knudsen et al., 2023*). Thus, there are a number of currently feasible steps for further extending the present findings.

While we did not set out to test predictive coding or other computational models in the current study, certain computationally relevant theories provide useful frameworks for interpreting the observed dysgranular co-activation during each imaging task. These include the Embodied Predictive Interoceptive Coding (EPIC) model (*Barrett and Simmons, 2015*), describing laminar-specific hypotheses for the interactions between anterior-mediated interoceptive predictions and posterior-mediated viscerosensation across the anteroposterior axis of the insula, and the Insula Hierarchical Modular Adaptive Interoception Control model (*Fermin et al., 2022*) using the hierarchical tripartite subdivision of the insula and describing parallel networks within prefrontal cortical and striatal regions that underpin higher-order interoceptive representation. For additional theoretical discussions of the computational functional neuroanatomy of interoception, see *Allen et al., 2022*, *Paulus et al., 2019*, and *Petzschner et al., 2021*. Collectively, these theories can be interpreted to suggest a model of bidirectional signal flow along the anteroposterior axis of the insula: while the granular posterior subregions may be the first to receive ascending visceral input, the agranular anterior subregions continuously issue predictions regarding the anticipated states of such input (*Barrett and Simmons, 2015*). From this perspective, the dysgranular mid-insula is well-positioned between the agranular and granular poles to integrate the signals from each source. Computations within dysgranular layers can link incoming sensations with top-down expectations (as suggested in the EPIC model), effectively translating visceral signals into endogenously meaningful interoceptive prediction error signals, and providing the rest of the brain with an internal context for interpreting the current bodily state (*Seth and Friston, 2016*).

Applying computational frameworks for interoception to the present findings may also be useful for interpreting the broad agranular insula activation found only during periods of increased bodily expectancies (i.e. 'divergence'). This subregion's engagement during the Anticipation window and expansion throughout the Peak period of the Saline dose likely reflects an increase in predictive processing during these conditions, as both groups expected a potential perturbation and were subsequently required to update their predictions upon experiencing a physiologically quiescent state instead. The reduced spatial extent of this activation in the ADE versus HC group could plausibly stem from an overreliance on top-down predictions and perhaps a pruning of spatial activation in the associated regions, similar to the functional alterations of neural activity that have been observed following the acquisition of a motor skill (*Wiestler and Diedrichsen, 2013*). Such 'efficient' neural exchanges might even result from a chronic inability to adjust confidence in bodily input in relation to its ambiguity (i.e. increasing confidence in visceral signals during bodily perturbation), and could occur alongside maladaptive allostatic interoceptive adjustments described in psychiatric disorders (*Barrett et al., 2016*; *Smith et al., 2020*). Computational theories of interoception can also be applied to the observed co-activation within the dysgranular mid-insula (i.e. 'convergence'). Dysgranular convergence aligns with our proposed intersectional (as opposed to overpassing) role for signal processing in this region, and could represent the generation of prediction errors as inflows of isoproterenol-induced viscerosensation are compared to task-specific interoceptive predictions focused by the lens of directed attention. While internally-directed attention alone might selectively enhance cortical representations of the viscera across wide swaths of the insula, as suggested previously (*Avery et al., 2014*; *Pollatos et al., 2007*), these past studies focus narrowly on perceptions of the viscera during physiological quiescent states when visceral signals are relatively ambiguous. By manipulating cardiorespiratory signals during the ISO 2.0 mcg dose and eliciting unambiguous perceptual states, the current study effectively spotlights the dysgranular subregion's role in heightened interoceptive awareness and provides a neuroanatomical focus for further examinations of bi-directional interoceptive processing in this important region.

The right-hemispheric increase in mid-insula activity, as well as the hemispheric asymmetries and altered functional connectivity found in the ADE group, has several implications for an interoceptive processing role for the dysgranular insula. Previous authors have suggested a right-hemispheric dominance in interoceptive processing in healthy populations (*Craig, 2009*; *Schulz, 2016*), and have pointed towards the right anterior insula as the key hub for body awareness (*Craig, 2009*). While a rightward emphasis in the magnitude of insular activation was found during ISO-induced cardiorespiratory sensations across both groups, this activity was located more posteriorly, towards the mid-insula (similar to previous isoproterenol fMRI studies *Hassanpour et al., 2016*; *Hassanpour et al., 2018*), where it was correlated with cardiorespiratory sensation. While this finding supports a role for the mid-insula as an integrative hub for interoceptive awareness, it was in contrast to a lack of clear associations with physiological signals, emotional feelings such as anxiety or excitement, or trait measures of ADE severity, despite the evocation of greater state anxiety following perturbation in the ADE individuals. Probing the symptomatic relevance of these potential brain-behaviour relationships may call for more refined approaches to the study of interoceptive perturbation. One option might be to evaluate the intersecting influence of several forms of perturbation, by studying both exogenous approaches such as isoproterenol infusion, and endogenous approaches such as mental stressors (i.e. Stroop task *Gianaros et al., 2005*), physical stressors (i.e. Valsalva (*Song et al., 2018*) or lower body negative pressure (*Kimmerly et al., 2005*), or social stressors (i.e. Trier Social Stress task (*Allen et al., 2017*) within the same individuals.

Despite observing several right hemispheric associations across both groups, the ADE group demonstrated a clear leftward activation asymmetry and distinct activation patterns in the left dysgranular mid-insula across both the ISO and VIA tasks. These left insula abnormalities are compatible with the 'locus of disruption' identified by *Nord et al., 2021*, when conducting an fMRI meta-analysis of existing top-down or bottom-up interoception studies of various psychiatric populations. However, beyond identifying a plausible region of cortex, the precise mechanisms underlying this disruption remain unclear. Assuming an intersectional role involving intermingling of top-down and bottom-up streams, it seems plausible that numerous mechanisms could contribute to the aberrant processing identified in the current study. Such mechanisms include a reduced attentional enhancement of ascending visceral input (*Avery et al., 2014*; *Kerr et al., 2016*), and/or an inflexibility in adjusting

to the salience of interoceptive signals occurring during perturbation, as revealed by computational modeling (*Smith et al., 2020*). Reduced flexibility of neural processing may also explain the singular finding of altered connectivity between the right mid-insula and left middle frontal gyrus following the interoceptive tasks in the ADE group, as this frontal region is likely involved in attentional control including attention bias to threat or internal representations (*Henseler et al., 2011*; *Sylvester et al., 2003*; *White et al., 2016*). An inability to return this coupling to baseline following interoceptive engagement (or a shift in the opposing direction than HCs) could suggest a dysregulated attentional switching mechanism, particularly since the degree of connectivity change was associated with trait measures of anxiety and depression severity.

From an affective standpoint, the observed hemispheric divergence in interoceptive processing is worth considering in the context of the functional lateralization of emotion theories proposed within the insula. Such theories (*Craig, 2005*; *Davidson et al., 1999*; *Davidson et al., 2004*; *Picard et al., 2016*), supported by meta-analytic (*Murphy et al., 2003*) and human brain stimulation (*Oppenheimer et al., 1992*) findings, have attributed the left anterior insula to the energy-enriching behaviors of the parasympathetic nervous system, and the right anterior insula with the energy-expending ones of the sympathetic nervous system (*Oppenheimer et al., 1992*). The current finding of right-hemispheric dominance for conscious awareness of isoproterenol (an arousal-like, and thus sympathetic-emulating manipulation), as well as the increased signal intensity in the right anterior insula during the expectancy of such manipulation, may be considered aligned with this perspective. Correspondingly, activity within the right insula has been associated with conscious awareness of threat (*Critchley et al., 2002*), and mouse models have shown causal evidence for the right anterior insula in orchestrating stress responses to aversive visceral stimuli (*Wu et al., 2020*). However, assumptions of autonomic opponency are likely overly simplistic, and it is more appropriate to consider these systems as functionally coupled across spinal and vagal systems instead of anatomically distinct (*Jänig, 2006*; *Jänig et al., 2017*). In this manner, an admittedly speculative emotional lateralization account of functional coupling might posit that the right anterior insula generates interoceptive predictions of spinal efference and the left anterior insula for those of vagal efference; in turn, the respective dysgranular areas would then be specialized towards receiving and computing the prediction errors of these different inputs. However, since there does not appear to be a neuroanatomical lateralization for afferent tracts (i.e. spinal input does not have a preference for the right hemisphere, and vagal not for the left) (*Dum et al., 2009*; *Shipley and Sanders, 1982*), it seems more plausible that the contribution of interoceptive processing to emotion requires a carefully integrated balance of both autonomic systems across both hemispheres. Based on the present data, such a balance appears to be asymmetrically shifted in the setting of psychiatric disorders affecting the anxiety, depression, and disordered eating spectrums, particularly during modulations of interoceptive state.

There are several limitations and strengths of the present approach to consider. First, while the quantification of insula asymmetries was strengthened by the use of previously established task-specific pipelines and justified by cytoarchitectural hypotheses leveraging prominent theories about interoceptive neurocircuitry, several analysis steps differed between the tasks that could impact the observed results, including the use of a block versus event-related design and differences in the regression of physiological noise and motion correction. This pre-specified approach was intentional. It allowed confidence that the analysis metrics were sound (i.e. that a voxel counted as co-activated would have good task-validity in relation to the methods of previously published ISO and VIA studies). However, it also meant that there was variability within-individuals in how the data was handled for each task. Future approaches involving harmonized pre-processing steps across both tasks may be considered. Second, the present approach to determining 'convergence' was admittedly simplistic in relying on co-activation as a determinant. Applying more sophisticated analysis techniques, such as those amenable to individual-level pattern analyses (as opposed to group activation maps) would provide greater insight into the individual responses of each participant. Such approaches are worth considering as they are a core component of personalized evaluations applying functional neuroimaging to diagnostic validation or treatment selection (*Goldstein-Piekarski et al., 2022*). Third, while the use of tripartite insular subregions was appropriate to the level of resolution in our data, future investigations could employ high-resolution fMRI to directly test the laminar-specific hypotheses of the predictive coding theories we have discussed. Fourth, while the conceptual approach to the top-down/bottom-up construct may mask the complexity of the brain (*Rauss and Pourtois, 2013*) and the

process of attention (*Hommel et al., 2019*), the integration of this concept with several behaviorally relevant tasks in the same individuals offers a useful and informative heuristic in its application. Finally, while our results focus on the insula's role in cardiorespiratory awareness, it's important to note that the insula isn't the only brain area involved in sensing internal body states (*Berntson and Khalsa, 2021*; *Mayeli et al., 2023*). Future research should explore structural and functional relationships between the insula and the rest of the brain; for example, relationships between white matter tracts (*Ghaziri et al., 2017*; *Ghaziri et al., 2018*), gray matter volume (*Hatton et al., 2012*), and functional coupling (*Cauda et al., 2011*). To build this more complete picture of interoceptive processing, other sophisticated network-based and whole-brain analysis approaches will be required, including those oriented towards the development of clinically-predictive models for interoceptive symptoms in psychiatric disorders.

In conclusion, this study confirms the pivotal role of the dysgranular insula in the processing of top-down and bottom-up interoceptive inputs, and validates the dysgranular mid-insula as a possible locus of interoceptive disruption in psychiatric disorders.

## Materials and methods

### Participants

Seventy individuals with primary diagnoses of anxiety, depression, and/or eating disorders (ADE) and 57 healthy comparisons (HC) were included in the study. Diagnoses were based on DSM-5 criteria using the Mini-International Neuropsychiatric Interview (*Sheehan et al., 1998*). Comorbid anxiety and depression were allowed, but those with panic disorder were excluded to reduce the likelihood of isoproterenol-induced panic anxiety (*Pohl et al., 1988*) in the scanner. Participants with a primary diagnosis of anxiety were further required to have scores greater than seven for the GAD-7 (*Spitzer et al., 2006*) and greater than 10 for the Overall Anxiety Severity and Impairment Scale (OASIS) (*Campbell-Sills et al., 2009*), indicative of clinically significant anxiety levels. Participants with eating disorders were required to have met lifetime DSM-5 criteria for anorexia nervosa but were required to be weight restored as defined by a minimum body mass index of 18.5 or greater for at least 1 month to control for the well-described state effects of acute starvation on neural activation (*Wierenga et al., 2015*). We used several self-report scales to measure symptom severity across the sample due to our interest in transdiagnostic markers of pathology. Psychotropic medications were allowed as long as there was no change in dosage during the four weeks preceding the MRI. Finally, a medical evaluation including a 12-lead electrocardiogram read by a cardiologist ensured all participants were generally healthy, free of cardiorespiratory, endocrine, metabolic, or other illnesses, as well as free of major neurological disorders or other psychiatric disorders besides ADE. Additional exclusion criteria involved anything that would render the participant unfit for the MRI environment, such as inability to remove body piercings or implanted electrodes. The study was part of a larger project investigating interoception in anorexia nervosa and generalized anxiety disorder (ClinicalTrials.gov Identifier: NCT02615119). Additional recruitment, screening, and exclusion information can be found in *Teed et al., 2022*, which used a subset of the current sample for a separate analysis.

All study procedures were approved by the Western Institutional Review Board (WIRB#20170214), and all participants provided written informed consent before participation. All data were collected at the Laureate Institute for Brain Research in Tulsa, OK.

### Experimental design and statistical analysis

During a single scanning session, participants completed two fMRI tasks designed to preferentially assess intersecting aspects of cardiorespiratory interoception while measuring blood oxygen level-dependent (BOLD) signals (*Figure 1*). The Isoproterenol Infusion (ISO) task was always performed before the Visceral Interoceptive Attention (VIA) task, which we considered to preferentially emphasize the bottom-up or top-down processing of interoceptive signals, respectively. The fixed order of these tasks reflected the project's primary emphasis on acquiring the infusion data. Complete details of these paradigms are described in the following sections. Both tasks were immediately preceded by and followed by an 8 min eyes-open resting state scan. For the resting state scans, participants were instructed to remain as still as possible, to keep their eyes open and fixated on a white cross centered upon a dark screen, and to 'clear your mind and do not think about anything in particular.' To minimize

head motion during scanning, a layer of tape was gently placed over the participant's forehead to provide them with tactile feedback regarding the head position. Cardiac and respiratory activity was measured simultaneously at 40 Hz during all scans using a pulse oximeter and thoracic respiratory belt, respectively. Finally, participants completed a high-resolution anatomical MRI scan before any resting state or task-based imaging data was collected.

### Isoproterenol infusion task

Intravenous bolus infusions of the peripherally-acting beta-adrenergic agonist isoproterenol (ISO) were used to manipulate afferent interoceptive signals in a bottom-up fashion. Infusions of 2.0 micrograms (mcg), 0.5 mcg, or Saline were administered 60 s into each 4 min infusion scan as per (*Hassanpour et al., 2018*) and (*Teed et al., 2022*). Infusion delivery was randomized, double-blinded, and repeated twice for each dose for a total of six infusions (i.e. six infusion scans). Throughout each infusion scan, participants were instructed to focus their attention on the sensations from their heart and lungs while providing real-time, continuous intensity ratings using an MRI-compatible dial (Current Designs Inc) with their dominant hand, the rating of which was visible to them on the screen while scanning. The intensity rating scale ranged from 0 (none or normal) to 10 (most ever). Thus, this task engaged top-down attention towards cardiorespiratory sensations while parametrically perturbing the bottom-up stream in the ISO trials, with an absence of such perturbation in Saline trials. Following our lab's previous studies (*Hassanpour et al., 2018*; *Teed et al., 2022*) we split the time course of this task into pre-defined epochs of interest, and focused on the Anticipation period (60–80 s, directly following the infusion but before the ISO-induced rise in heart rate) and the Peak period (80–120 s, encompassing the onset and peak magnitude of ISO-induced heart rate change), and contrasted these with the Baseline period (the initial 45 s of the task). Following each scan, participants also retrospectively reported the intensity of perceived cardiac or respiratory sensations, separately, that they felt during the scan, as well as their anxiety or excitement. To familiarize participants with the task and isoproterenol-induced sensations, single-blinded infusions of 1 mcg ISO and Saline (one trial each) were given before the MRI session.

### Visceral interoception attention task

The visceral interoceptive attention (VIA) task was used to assess the top-down attentional processing of interoceptive signals at rest (i.e. no infusions were delivered). This minimalistic task, adapted from *Simmons et al., 2013*, involved repeated trials of interoceptive or exteroceptive attention delivered across three 6 min runs, with the order of these runs randomized across participants. During the interoceptive attention trials, the words 'HEART' 'LUNGS,' or 'STOMACH' were presented on the screen for 8 s, during which participants were instructed to focus their attention on the naturally-occurring sensations experienced from that organ. During the exteroceptive attention control trials, the word 'TARGET' was shown on the screen at changing shades of black to gray, and participants were instructed to pay attention to the degree of color change. After half of all trials in each condition, participants were asked to indicate via an MRI-compatible button box the intensity of sensations they had perceived from that organ (interoceptive condition) or the intensity of the color change (exteroceptive condition) on a rating scale of 1 (no sensation/no change in color) to 7 (high intensity).

### MRI data acquisition

All scans were conducted in a three Tesla General Electric scanner with an eight-channel head coil (GE MR750; GE Healthcare). Anatomical images were acquired via T1-weighted, magnetization-prepared, rapid gradient-echo sequences images (MPRAGE). MPRAGE parameters were: 190 axial slices, slice thickness = 0.9 mm, TR/TE = 5/2.012 ms, FOV = 240 × 192 mm$^2$, matrix size 256 × 256, flip angle 8 degrees, inversion time 725 ms, SENSE acceleration $R$=2, with a sampling bandwidth of 31.2. For the task-based and resting state functional scans, T2*-weighted images were acquired via an echoplanar sequence of the following parameters: single-shot gradient echo planar imaging (EPI) sequence covering the whole brain, obtaining 39 axial slices, 2.9 mm thick with no gap and a voxel size of 1.875 × 1.875 × 2.9 mm$^3$, with TR/TE = 2000/27 ms, FOV = 240 × 240 mm$^2$, flip angle 78 degrees, SENSE acceleration $R$=2 with a 96 × 96 matrix.

## fMRI task-based analysis

Each task was analyzed separately in AFNI (*Cox, 1996*), following the previously-published preprocessing pipelines specific to each task (*Simmons et al., 2013*; *Teed et al., 2022*). Preprocessing steps common to both tasks included removal of the first four volumes (ISO) or five volumes (VIA) for steady-state tissue magnetization, despiking, slice time correction, spatial smoothing (6 mm full width at half maximum Gaussian kernel), co-registration to the first volume, scaling of each voxel, and normalization to Talairach space. In the ISO task, temporal fluctuations of cardiac and respiratory frequencies and their first-order harmonics were additionally removed from the imaging data using RETROICOR, implemented via custom code in MATLAB (MathWorks).

After preprocessing, whole-brain individual-level activation maps were created. For the ISO task, this involved regressing out baseline drift and contrasting average BOLD percent signal change (PSC) within each epoch of interest (Anticipation and Peak, separately) against the average PSC during the Baseline period. For the VIA task, this involved a multiple linear regression model that included terms for baseline drift, six motion parameters, and each interoceptive and exteroceptive attention condition, and contrasting PSC associated with cardiorespiratory attention (i.e. collapsed across 'Heart' and 'Lung' trials) against that of exteroceptive attention ('Target' control condition). This cardiorespiratory contrast was specifically chosen as it best matched the cardiorespiratory perturbation induced by ISO.

After the creation of individual-level maps, quality control checks were performed. Participants were excluded from further group analysis if they had poor-quality imaging data according to the previously established approaches in each task. For the ISO task, individual runs were excluded if more than 20% of all time points had over 0.3 mm motion, based on the Euclidean norm of the six motion parameters; following this, participants were excluded who were missing both runs of the Saline or 2.0 mcg doses. For the VIA task, all time points with an outlier fraction greater than 5% or an estimated motion greater than 0.3 mm were censored, and any run in which this censoring removed over 20% of volumes was excluded. If a participant was missing two or more VIA runs (out of three), they were removed from the analysis. To avoid an unbalanced conjunction analysis in the following step, participants excluded from one task were also removed from the other task. The groups were then propensity matched based on age, sex, and BMI to equalize the sample sizes (*MatchIt* function in R), resulting in a final sample of participants who had high-quality imaging data in *both* tasks.

Using this propensity-matched sample, group-level statistical activation maps were created for each task and group separately, using whole-brain voxel-wise t-tests (*3dttest ++*in AFNI) with Equitable Threshold And Clustering corrections for an overall False Positive Rate of 5% (*Cox, 2019*). With these group activation maps, we were thus able to identify voxels significantly active at the group level in each experimental condition.

## Convergence maps

To examine voxels co-activated across the ISO and VIA tasks, we restricted the functional neuroimaging analysis to the specific portions of each task that best exemplified the interoceptive processes of interest: namely, Heart + Lung interoceptive attention relative to exteroceptive attention in the VIA task, and the Peak period of the 2.0 mcg dose relative to the Baseline period in the ISO task. We narrowed the focus of analysis to this specific time period and dose of the ISO task because it represents, as shown in previous studies, the peak level of cardiorespiratory perturbation that can be confidently ascertained by subjective reports in all individuals (*Hassanpour et al., 2018*; *Teed et al., 2022*). Using the statistically-thresholded group activation maps from each task, binarized into active and non-active voxels, we created convergence maps by joining the binary maps across the specified contrasts of each task. These convergence maps thus identified which voxels were commonly activated, i.e., *convergent*, at the group level during both tasks. Due to our pre-specified focus on cytoarchitectural processing of the insular cortex, we subdivided these convergence maps (after converting to MNI space) into three bilateral insular subregions of interest, defined using the Brainnetome probabilistic cytoarchitectonic atlas (*Fan et al., 2016*), as follows: agranular anterior (regions 165 and 167 for the left hemisphere, 166 and 168 for the right), dysgranular mid (regions 169 and 173 for the left hemisphere, 170 and 174 for the right), and granular posterior (region 163 and 171 for the left hemisphere, 164 and 172 for the right).

Each convergence map was evaluated in four ways: (1) the spatial extent of convergence (i.e. number of convergent voxels) within each insular subregion, tested for group differences via

chi-square tests, (2) the spatial similarity of patterns of convergence between HC versus ADE groups, measured via Dice coefficient separately for each hemisphere (using AFNI's *3ddot*), (3) the degree of task overlap within each group and hemisphere separately, quantified via the overlap coefficient (i.e. the proportion of the ISO cluster overlapping with the VIA cluster, as per *Bowring et al., 2019*), and finally (4) the average PSC across convergent voxels within each cytoarchitectonic subregion, compared across groups and hemispheres using linear mixed effects regression (*lmerTest* library in R, with the following equation: percent signal change ~group * hemisphere + (1|subject)). In this linear model, the fixed effects of group and hemisphere were tested with respect to the magnitude of activation within subregion-specific convergent voxels, and significant main effects were interpreted using post-hoc contrasts between the estimated marginal means of each condition (*emmeans* library in R). Additionally, linear regression was used to examine possible associations between PSC in the hemisphere- and subregion-specific convergent voxels and the following variables: trait measures of anxiety and depression, the objective degree of peripheral perturbation (i.e. mean change in heart-rate during the Peak period of the 2.0 mcg dose), real-time subjective rating of cardiorespiratory intensity (i.e. mean change in dial rating during the Peak period of the 2.0 mcg dose), and retrospective ratings of cardiac and respiratory intensity, anxiety, and excitement following the 2.0 mcg dose. For these associations, we used a dimensional approach that collapsed across groups, considering all participants on a continuous scale.

## fMRI resting state analysis

In an exploratory whole-brain resting state functional connectivity (FC) analysis, we aimed to identify which connections, if any, between convergent regions and the rest of the brain were altered following the interoceptive manipulations, and if these patterns of connectivity change differed between groups. We predicted that residual group-specific changes in functional connectivity of the insular regions activated during the ISO and VIA tasks would be present in the resting state scan at the end of the scanning session, reflecting additional neural signatures of interoceptive processing. We did not include a temporal control condition for this exploratory analysis, and thus we were unable to examine the specificity of insular connectivity changes across time in relation to test-retest reliability measures of the group's resting state data.

Resting state scans were analyzed in AFNI with a preprocessing pipeline optimized for connectivity analysis. After discarding the first five images to ensure steady-state tissue magnetization, we removed spikes in the signal intensity time course, and removed physiological noise using respiration volume per time (RVT) (*Birn et al., 2008*) and RETROICOR (*Glover et al., 2000*) regressors that model slow blood oxygenation level fluctuations and time-locked cardiac and respiratory artifacts, respectively. The RVT regressors consisted of the RVT function and four delayed terms at 5, 10, 15, and 20 s, and the RETROICOR regressors consisted of four respiratory and four cardiac generated on a slice-wise basis by AFNI's 'RetroTS.m' script (*Jo et al., 2010*). Further preprocessing steps included slice-timing correction, rigid body realignment, warping to MNI space, and smoothing with a 6 mm full-width half-maximum Gaussian kernel. Next, we applied a high-pass filter to remove all non-relevant ultra-slow fluctuations below 0.01 Hz, and then further reduced potential contributions of non-neural sources by regressing out the following nuisance variables: (1) 12 head motion regressors (6 realignment parameters and their derivatives), (2) voxel-wise local white matter regressors, and (3) three principal components of ventricle signals (ANATICOR). Following preprocessing, any subject with greater than 10% of volumes above our motion cut-off of 0.3 mm for the Euclidean norm of the motion derivatives was excluded from further analysis.

From the pre- and post-interoception resting state scans, time series were extracted from the convergent ROIs within the left and right hemispheres for HC and ADE groups separately, and correlated these respective time series against those of all other voxels in the brain to create an FC map for each resting state period (i.e. pre- and post-interoception). The resulting correlation coefficients were converted to Fisher z-statistics to produce a more normally distributed variable. To examine group differences in the effect of time (i.e. pre- versus post-interoception tasks) on resting-state FC, all resulting clusters of this group*time interaction analysis were corrected for multiple comparisons using the *3dClustSim* function in AFNI, using 10,000 Monte Carlo simulations of whole-brain fMRI data to determine the cluster size at which the false positive probability was below $\alpha<0.05$ with an uncorrected voxel-wise threshold of p<0.001, yielding a threshold of 150 contiguous voxels.

This approach allowed the identification of regions that changed connectivity with the convergent insula differently for HC vs. ADE groups following top-down and bottom-up interoception. Finally, we conducted a correlational analysis to examine whether individual changes in FC were related to trait measures of anxiety and depression in the transdiagnostic sample, using the Bonferroni method for multiple comparison correction. These correlations were performed to help interpret the functional significance of residual changes, based on previous studies showing trait-related differences following interoception (*De la Cruz et al., 2023*).

### Divergence maps

To isolate voxels selective to the engagement of top-down expectancies about the body, we focused on specific time windows of the ISO task were where expectations of potential interoceptive change were preferentially engaged: namely, the Anticipation period across all doses, and the Peak period of the Saline dose. The Anticipation period covered the 20 s epoch right after the intravenous infusion was delivered but before the onset of any cardiorespiratory perturbation (in the case of an isoproterenol dose administration). Thus, the Anticipation period represented a time when no peripheral change was occurring, but also one where participants were unsure about whether or not such a change was coming. This state of uncertainty would continue into the Peak period of the Saline dose, where again, no peripheral physiological change was induced. In the setting of isoproterenol dose administration, this state of uncertainty would resolve when participants correctly identified and rated the onset of changes in cardiorespiratory sensation. Based on previous studies from our lab using this task, we expected that by the end of the Saline Peak window, some participants would update their beliefs about what had occurred after realizing that they had not reported experiencing a cardiorespiratory perturbation (i.e. their dial ratings remained at zero). Together, the Anticipation period and the ensuing Peak period of the Saline dose share some similarities with the VIA task in their involvement of top-down interoceptive attention during physiological rest, but critically, they differ in their added expectancies about possible changes in the state of the body. Thus, by isolating voxels preferentially active during these ISO anticipatory windows that were not active during VIA top-down attention alone, it we presumed it would be possible to separate the effect of bodily expectancies from the general effect of bodily attention.

For this aim, group-level activation maps from the Anticipation period (averaged across all doses) and the Peak period of the Saline dose, both relative to the Baseline period, were examined. Like the convergence map approach described above, these maps were binarized within each period and group separately, transformed to MNI space, reduced to the cytoarchitecturally-subdivided insular cortex, and joined with the binarized activation map for the Heart and Lung relative to Target contrast in the VIA task. Here, our focus was instead on the *divergent* voxels, i.e., on those that were active only during specific periods of the ISO task and that were not co-active during the VIA task. As per our hypothesis regarding anterior insula involvement in interoceptive expectancy processing, these 'ISO task only' voxels were restricted to those within the agranular anterior insula, and were evaluated in two ways: (1) spatial extent of activation (i.e. number of voxels) across groups, time periods, and hemispheres, examining differences via chi-square tests with Bonferroni correction for multiple comparisons, and (2) magnitude of activation, examining differences in average PSC across the fixed effects of group, hemisphere, and time period using linear mixed effects regression and interpreting significant main effects using post-hoc contrasts of estimated marginal means (as previously described). Finally, we examined possible associations between average PSC in these expectancy-specific agranular regions and trait measures of anxiety using linear regression, transdiagnostically across groups.

### Physiological and behavioral analysis

To test for group differences in the time courses of heart rate and real-time dial ratings during the ISO task (2.0mcg isoproterenol and Saline infusions), we used cluster-based permutation testing (*permutes* package in R) separately for each dose after averaging across both trials. A significant group difference was defined as any contiguous set of time points (two or more continuous seconds) where permutation testing revealed a statistical difference at $p < 0.05$. We used linear regression transdiagnostically to test if the magnitude of heart rate or dial rating change, when averaged across each epoch of interest, was associated with dimensional measures of trait anxiety and depression across groups. Finally, we used two-tailed t-tests separately for each task and condition to test group

differences in retrospective ratings of interoceptive intensity, anxiety, and excitement in the ISO task and interoceptive and exteroceptive intensity in the VIA task. This approach reflected our primary interest in investigating group differences in perceptual ratings, rather than performing additional within-group comparisons across modalities or doses.

## Acknowledgements

The authors would like to thank Maria Puhl and Gabriel Morrison for their contributions to data preprocessing, as well as Valerie Upshaw and Rachel Lapidus for their assistance with data collection.

## Additional information

### Funding

| Funder | Grant reference number | Author |
| --- | --- | --- |
| National Institute of Mental Health | K23MH112949 | Sahib Khalsa |
| National Institute of Mental Health | R01MH127225 | Sahib Khalsa |
| National Institute of General Medical Sciences | 1P20GM121312 | Sahib Khalsa |

The funders had no role in study design, data collection and interpretation, or the decision to submit the work for publication.

### Author contributions

Emily M Adamic, Data curation, Formal analysis, Validation, Visualization, Methodology, Writing – original draft, Writing – review and editing; Adam R Teed, Data curation, Formal analysis, Methodology, Writing – review and editing; Jason Avery, Feliberto de la Cruz, Formal analysis, Validation, Methodology, Writing – review and editing; Sahib Khalsa, Conceptualization, Resources, Data curation, Formal analysis, Supervision, Funding acquisition, Investigation, Visualization, Methodology, Writing – original draft, Project administration, Writing – review and editing

### Author ORCIDs

Sahib Khalsa [ID] https://orcid.org/0000-0003-2124-8585

### Ethics

Clinical trial registration Clinicaltrials.gov NCT02615119.
Human subjects: All study procedures were approved by the Western Institutional Review Board, and all participants provided written informed consent before participation. (WIRB#20170214).
All study procedures were approved by the Western Institutional Review Board (Western IRB #20170214), and all participants provided written informed consent before participation.

Reviewer #2 (Public Review): https://doi.org/10.7554/eLife.92820.3.sa1
Reviewer #3 (Public Review): https://doi.org/10.7554/eLife.92820.3.sa2
Reviewer #4 (Public Review): https://doi.org/10.7554/eLife.92820.3.sa3
Author response https://doi.org/10.7554/eLife.92820.3.sa4

## Additional files

### Supplementary files
• MDAR checklist

## Data availability

The raw data for Figures 3 to 7 are posted on Dryad. Custom AFNI and R scripts used to generate these figures are posted on the Open Science Framework preregistration for this project (*Adamic et al., 2021*).

The following dataset was generated:

| Author(s) | Year | Dataset title | Dataset URL | Database and Identifier |
|---|---|---|---|---|
| Khalsa et al. | 2024 | Hemispheric divergence of interoceptive processing across psychiatric disorders | https://dx.doi.org/10.5061/dryad.xgxd254rc | Dryad Digital Repository, 10.5061/dryad.xgxd254rc |

The following previously published dataset was used:

| Author(s) | Year | Dataset title | Dataset URL | Database and Identifier |
|---|---|---|---|---|
| Adamic E, Avery J, Khalsa SS, Teed AR | 2021 | Convergence between goal-directed and stimulus-driven streams of cardiorespiratory interoception | https://doi.org/10.17605/OSF.IO/6NXA3 | Open Science Framework, 10.17605/OSF.IO/6NXA3 |

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
