## [Editor Report · eLife assessment]

This **fundamental** study provides **compelling** evidence for dysgranular insular involvement in top-down and bottom-up interoceptive processing by building on previous evidence using state-of-the-art methods. Its translational application in ADE patients corroborates the assumption that the mid-insula may indeed be a locus of 'interoceptive disruption' in psychiatric disorders, which underscores the study's high relevance for both body-brain as well as clinical research.

---

## [Referee Report · Reviewer #2 (Public Review)]

Summary:

The authors have conducted an exceptionally informative series of studies investigating the neural basis of interoception in transdiagnostic psychiatric symptoms. By comparing differential and overlapping neural activation during 'top-down' and 'bottom-up' interoceptive tasks, they reveal convergent activation largely localised to the ventral dysgranular subregion ('mid-insula'), which differs in extent between patients and controls, replicating and extending previous suggestions of this region as a central locus of disruption in psychiatric disorders. Their work also reveals different extents of divergent activation in the anterior insula during anticipation of interoceptive disruption. This substantially advances our previous knowledge of the anatomy of interoception, and confirms theoretical predictions of the roles of different cytoarchitectural subregions of the insula in interoceptive dysfunction in mental health conditions.

Strengths:

The work is exceptional in terms of breadth and depth, making use of multiple imaging and analysis techniques which are non-standard and go well beyond what is known today. The study is statistically well-powered and the tasks are well-validated in the literature. To my knowledge, these functions of the insula in interoception and mental health have never been compared directly before, so the results are novel and informative for both basic science and psychiatry. The work is strongly theory-driven, building on and directly testing results from influential theories and previous studies. It is likely that the results will strengthen our theoretical models of interoception and advance psychiatric studies of the insula.

Weaknesses:

The study has three limitations. (1) The interpretation of the resting-state isoproterenol data could potentially represent fluctuations over time rather than following interoception specifically; future studies should investigate test-retest reliability of this measure. Note this does not preclude the strong conclusions which can be drawn from the authors' task-based data. (2) The transdiagnostic patient sample was almost entirely female, and many were currently taking psychotropic medications; future studies should replicate these effects in unmedicated, sex-balanced samples (3) As the authors point out, there may have been task-specific preprocessing/analysis differences that influenced results, for example due to physiological correction in one but not both tasks; however, there are also merits to this analysis approach, such as comparability with previous studies.

---

## [Referee Report · Reviewer #3 (Public Review)]

Summary:

Adamic and colleagues present fMRI data from ADE patients and a healthy control group acquired during two interoceptive tasks (attention and perturbation) from the same session. They report convergent activity within the granular and dysgranular insular cortex during both tasks, with a patient group-specific lateralisation effect. Furthermore, insular functional connectivity was found to be linked to disease severity.

Strengths:

The study is well-designed and - despite some limitations noted by the authors - provides much-needed insight into the functional pathways of interoceptive processing in health and disease. The manuscript is clear, concise, and well-written.

Weaknesses:

None remain after the authors' revision.

---

## [Referee Report · Reviewer #4 (Public Review)]

Summary:

In the manuscript titled "Hemispheric Divergence of Interoceptive Processing Across Psychiatric Disorders", the authors analyzed a subset of data collected for a larger project investigating interoception in anorexia nervosa and generalized anxiety disorder (ClinicalTrials.gov Identifier: NCT02615119). This study utilized fMRI and various analyses with a special focus on the insula and its connectivity to map the neural commonalities and differences in both top-down and bottom-up interoceptive processing.

The primary aim was to compare whether these neural activations were quantitatively and qualitatively different in a sample of healthy controls (HC) versus patients diagnosed with anxiety, depression, and/or eating disorders (ADE).

The study initially recruited 70 patients with primary diagnoses of ADE and 57 HC. After applying exclusion criteria, the final sample consisted of 46 ADE patients and 46 matched HC. Participants underwent task-related and resting-state fMRI scan sessions.

Specifically, participants performed 2 tasks in fMRI: (i) a bottom-up interoceptive (ISO) task involving intravenous infusions of isoproterenol (a peripherally-acting beta-adrenergic receptor agonist) administered in a double-blind, placebo-controlled fashion to alter cardiovascular activity where participants were asked about their visceral awareness; and (ii) a top-down interoceptive attention (VIA) task where participants were asked to focus on their visceral sensations triggered by words indicating specific body parts (e.g., STOMACH, HEART, LUNGS) or to pay attention to color changes of the word TARGET during an exteroceptive control task.

Main results show overlapping patterns of neural activation within the dysgranular mid-insula during top-down and bottom-up interoceptive processing with hemispheric differences. The patterns of dysgranular activation distinguished individuals with ADE compared to HC. Also differences in the activation of the anterior agranular insula during periods of interoceptive uncertainty differentiate ADE patients from HC.

Strengths:

- This is a very nice study that aligns with modern Clinical Neuroscience approaches, as recommended by NIH policy (i.e. RDoC initiative), which puts emphasis describing clinical conditions via transdiagnostic dimensions measured on psychological processes, behaviors, and neural processes rather than merely identifying a series of symptoms.

I appreciated very much the different analyses that authors performed to characterize differences at the qualitative and quantitative regarding the insular activity and its connectivity during bottom-up and top-down interoceptive processes.

These findings may open avenues for new studies that will explain the mechanisms underlying these phenomena and provide useful insights for developing novel interventions.

Weaknesses:

Weakness/Requests of additional clarifications

(1) The sample

(1.1) The authors describe the patient's group as having a primary diagnosis of anxiety, depression, and/or eating disorders. However, Table 1 shows that the majority had Anxiety disorders, some Major Depression (it is not clear which are the percentages of patients that at the time of the study had a concurred problem of major depression, please clarify), and very few had a diagnosis of Anorexia Nervosa. The leftward activation asymmetry and distinct activation patterns in the left dysgranular mid-insula across both the ISO and VIA tasks found on ADE did not correlate with symptoms measured by the SCOFF questionnaire, but correlated with anxiety and depressive symptoms. It would be nice if the authors can comment on these results in relation to eating disorders.

(1.2) Furthermore, the sample consisted of 5 males and 41 females in the HC group and 1 male and 45 females in the ADE group. In order to generalize these findings, the authors should acknowledge this gender imbalance and discuss whether they expect similar results in a predominantly male sample.

(2) The procedure

While the fixed order of tasks reflects the primary emphasis on acquiring data from the infusion (ISO) task, this could introduce confounding order effects. The authors should acknowledge this as a limitation of this study.

(3) The rationale behind the study

- The authors recognized that there was a broader aim behind this data collection. It would be important to clarify a little bit more how the differences in insular areas mapping both (or specifically) bottom-up and top-down interoceptive processes and insular connectivity, recorded in ADE patients compared to healthy controls (HC), contribute to psychiatric diagnoses (hypothesis 3).

For example, they should explain the psychopathological dimensions common to the three patient groups. Are disturbances in bottom-up and top-down interoceptive processing common traits in these patients, reflected in the asymmetric interhemispheric dysgranular mid-insular activation? The link between these disturbances and anatomical evidence of convergence/divergence of top-down vs. bottom-up interoceptive processes should be clearly stated.

(4) Operationalization of Convergence / Divergence maps underlying top-down and bottom-up interoceptive processes in HC vs ADE patients

It is not clear to me the concept of Convergence / Divergence maps underlying top-down and bottom-up interoceptive processes. The authors want to compare, in HCs and ADE patients, the neural structures that are co-activated (convergence maps) vs those that are uniquely involved (divergence maps) in top-down and bottom-up interoceptive processes in the two groups. Thus, I would expect that these two different analyses would have been performed on similar portions of data, instead different moments of the tasks (=different bottom-up / top-down interoceptive processes) have been analyzed.

Specifically, the convergence maps have been identified by comparing active voxels recorded when participants were focusing on the heart and the lungs (compared to when they were focused on the exteroceptive features of the target) in the VIA task, and during infusions (Peak period) of 2mcg isoproterenol (compared to baseline) in the ISO task. The divergence maps have been identified by comparing voxels uniquely active during the anticipatory phases of both isoproterenol and saline infusions (compared to baseline) and during the peak period of saline dose of the ISO task with respect to when participants focused their attention on the heart and the lungs (compared to when they were focuses on the exteroceptive features) in the VIA task.

I understand the idea of mapping interoceptive uncertainty, however I think that these two analyses do not show commonalities and differences in the neural structures involved in bottom up vs top down processes (in ADE vs HC), but also neural correlates underlying different types of interoceptive processes involving or nor top-down expectations.

According to the authors, which is the most important neural marker that differentiates the ADE group: the difference in hemispheric activations within the left and right dysgranular insula or the less granular anterior insular activation during periods of interoceptive uncertainty? Also, do they reflect different transdiagnostic dimensions?

(5) Collected physiological measures

The authors speak about cardiorespiratory interoceptive processes, but they only included cardiac measures. Including respiratory changes could provide a more comprehensive comparison between bottom-up signals and top-down attentional processes. Also, I guess that the "STOMACH" trials of the VIA task were not analyzed in this study since those are used in the bigger study and since no gastric measures were collected? Please clarify this point.

(6) ISO task instructions

To better understand the task and participants' expectations, could the authors clarify the instructions given to participants regarding the isoproterenol and saline infusions. Did the participants have two types of expectations?

(7) Title of the study

I understand that the term "divergence" in the title refers to the different hemispheric activations characterizing ADE patients compared to HC. However, it also suggests an analysis based on convergence/divergence maps, which might be ambiguous. Could the authors make some small modifications to the title to make it clearer?

(8) Caption of Figure 7

The caption of Fig.7 notes that no difference in HR was found during the Saline infusion between the HC and ADE groups. However, it would be fair to mention the significant difference in dial ratings observed during the Saline infusion. How do the authors explain this difference?

Typos

Figure 3 In Figure 3, "Hemispheric divergence", I think, should be corrected to "Hemispheric convergence."

I believe that by addressing these points, the manuscript will provide a clearer and more comprehensive understanding of the rationale, methods, and findings underlying this study.

---

## [Author Response]

The following is the authors’ response to the original reviews.

**Reviewer 1:**
One concern is regarding the experimental task design. Currently, only subjective reports of interoceptive intensity are taken into account, the addition of objective behavioural measures would have given additional value to the study and its impact.

To address this comment, we calculated interoceptive accuracy during the cardiorespiratory perturbation (isoproterenol) task according to our previous methods (e.g., Khalsa et al 2009 Int J Psychophys, Khalsa et al, 2015 IJED, Khalsa et al 2020 Psychophys, Hassanpour et al, 2018 NPP, Teed et al 2022 JAMA Psych). Thus, we quantified interoceptive accuracy as the cross-correlation between heart rate and real-time cardiorespiratory perception; specifically, the zero-lag cross-correlation between the heart rate and dial rating time series, and the maximum cross-correlation between these time series while allowing for different temporal delays (or lags). As expected, we found a dose-related increase in interoceptive accuracy from the 0.5mcg moderate perturbation dose (for which neuroimaging maps were not included in the current study) to the 2.0mcg high perturbation dose: zero-lag cross-correlations of 0.25 and 0.61, maximum cross-correlations of 0.41 and 0.73, for 0.5mcg and 2.0mcg doses, respectively, when averaged across all participants in the current study. Taking a closer examination at just the 2.0mcg dose, there were no group differences in zero-lag cross-correlation (t89=-0.68, p=0.50) or maximum cross-correlation (t87=-1.0, p=0.32) (depicted below, panel A). Furthermore, there were no associations between either of these interoceptive accuracy measures and the magnitude of activation within bilateral dysgranular convergent regions (F1=0.27 and 0.01, p=0.61 and 0.91, for the main effect of percent signal change on max and zero-lag cross-correlations, respectively; depicted below, panel B). When considering the significant correlation between the right insula signal intensity and subjective dial ratings, this lack of association with interoceptive accuracy suggests that the right dysgranular convergent insula was preferentially tracking the magnitude estimation rather than accuracy facet of interoceptive awareness during cardiorespiratory perturbation. Notably, during the saline placebo infusion, there were no systematic changes in heart rate and thus no systematic change in dial rating, precluding the calculation of the cross-correlation as a measure of interoceptive accuracy.

In reviewing these findings, we did not feel that the results add meaningful information to our interpretation of convergence, and accordingly we have chosen not to include it in the manuscript.

**Author response image 1. sa4fig1:** (**A**) Interoceptive accuracy during 2.0mcg isoproterenol perturbation, as measured by the maximum (left panel) and zero-lag (right panel) cross-correlation between the time series of heart rate and perceptual dial rating. There were no differences between groups. (**B**) There were no associations between interoceptive accuracy ratings and signal intensity within the convergence dysgranular insula during the Peak period of 2.0mcg perturbation.

This brings me to my second concern. The authors mostly refer to their own previous work, without highlighting other methods used in the field. Some tasks measure interoceptive accuracy or other behavioural outcomes, instead of merely subjective intensity. Expanding the scientific context would aid the understanding and integration of this study with the rest of the field.

Given our focus on the neural basis of bottom-up perturbations of interoception, we found it relevant to reference previous studies from our lab, as we built directly upon these previous findings to inform the hypotheses and design of the current experiment, but we can appreciate to provide a broader view of the literature. To expand the contextual frame, we have cited two fMRI meta-analyses of cardiac and gastrointestinal interoception (line 101). There are few studies that have used comparable perturbation approaches during neuroimaging in clinical populations, although we have referenced an exemplar study from the respiratory domain by Harrison et al (2021) in the discussion (line 612). In considering this comment more carefully, we felt that expanding the context further to other task-based methods or behavioral outcomes would shift the focus beyond our emphasis on the insular cortex and top-down/bottom-up convergence, though we have previously discussed and integrated such approaches (e.g., Khalsa & Lapidus, 2016 Front Psych, Khalsa et al, 2018 Biol Psychiatry CNNI, Khalsa et al 2022, Curr Psych Rep).

Lastly, the suggestions for future research lack substance compared to the richness of the discussion. I recommend a slight revision of the introduction/discussion. There is text in the discussion (explanatory or illuminating) which is better suited to the introduction.

When discussing our study limitations (beginning line 732), we offer numerous areas for future research including different preprocessing pipelines, more sophisticated analysis techniques (such as multivariate pattern analysis) that would allow for individual-level inferences regarding convergent patterns of activation within the insula. However, we have revised the last sentence of our limitations paragraph (line 757), and have added more specificity regarding future approaches examining insular and whole-brain interoceptive signal flow.

**Reviewer 2:**
(1) The interpretation of the resting-state data is not quite as clear-cut as the task-based data - as presented currently, changes could potentially represent fluctuations over time rather than following interoception specifically. In contrast, much stronger conclusions can be drawn from the authors' task-based data. …I was also unsure about the interpretation of the resting state analysis (Figure 5), as there was no control condition without interoceptive tasks, meaning any change could represent a change over time that differed between groups and not necessarily a change from pre- to post-interoception. Relatedly I wondered if the authors had calculated the test-retest reliability of the resting state data (e.g. intraclass correlation coefficients for the whole-brain functional connective of convergent dysgranular insula subregions and left middle frontal gyrus before vs. after the tasks), as it would be generally useful for the field to know its stability.

We have acknowledged the lack of a control condition in the isoproterenol task (note that the VIA task contained an exteroceptive trial that was included in the brain image contrast analysis). We have also provided further justification for our approach in both the Methods (see the first paragraph “fMRI resting state analysis” subsection) and Results (see the last paragraph of the “Convergence analysis” subsection). We cannot estimate test-retest reliability from the current dataset, given that we do not have resting state scans separated by a similar time frame without the performance of the interoceptive tasks in between (this is now clarified in line 346).

(2) The transdiagnostic sample could be better characterised in terms of diagnostic information, and was almost entirely female; it is also unclear what the effect of psychotropic medications may have been on the results given the effects of (e.g.) serotonergic medication on the BOLD signal. …Table 1 would be substantially improved by a fuller clinical characterisation of the specific sample included in the analysis - the diagnostic acronyms included in the table caption are not used in the table itself at present and would be an excellent addition, describing, for example, the demographics and symptom scores of patients meeting criteria for MDD, GAD, and AN (and perhaps those meeting criteria for more than 1). Similarly, additional information about the specific medications patients (or controls?) were taking in this study would be welcome (given the potential influences of common medications (e.g. antidepressants) on neurovascular coupling).

We have expanded Table 1 to include more specific diagnostic information for the transdiagnostic ADE group (GAD, MDD, and/or AN, as well as other psychiatric diagnoses). We have also included medication use.

Finally, Figures 7c and 7d would be greatly improved by showing individual data points if possible, and there may be a typo in the caption 'The cardiac group reported higher cardiac intensity ratings in the ADE group'.

We have adjusted Figure 7c and 7d to include individual data points, as we agree that this provides greater transparency to the data itself. We have also fixed the typo in the figure caption.

(3) As the authors point out, there may have been task-specific preprocessing/analysis differences that influenced results, for example, due to physiological correction in one but not both tasks. Although I note this is mentioned in the limitations, it was not clear to me why physiological noise was removed from the ISO task and whether it would be possible to do the same in the VIA task, which could be important for the most robust comparison of the two.

In this study, we intentionally chose different task-specific preprocessing pipelines so we could ensure that our results were not simply due to new ways of handling the data. This would allow us to evaluate evidence of replicating the previous group-level findings of insular activation that informed the current approach and hypotheses. We agree that a harmonized approach is also merited, and in a subsequent project using this dataset, we have matched preprocessing pipelines for a connectivity-based analysis, to best facilitate comparison across tasks. We look forward to sharing those results with the scientific community in due time.

**Reviewer 3:**
Maybe I missed it (and my apologies in case I did), but there were a few instances where it was not entirely clear whether differential effects (say between groups or conditions) were compared directly, as would be required. One example is l. 459 ff: The authors report the interesting lateralisation effect for the two interception tasks and say it was absent in the exteroceptive VIA task. As a reader, it would be great to know whether that finding (effect in one condition but not in the other) is meaningful, i.e. whether the direct comparison becomes statistically significant. … The same applies to later comparisons, for example, the correlations reported in l. 465 ff (do these differ from one another?) as well as the FC patterns reported in l. 476 ff - again, there is a specific increase in the ADE group (but not in the HC), but is this between-group difference statistically meaningful?

Thank you for these questions. We have added greater detail in the Results section in order to increase clarity regarding which statistical comparisons support which conclusions. Generally, we limited our comparisons to the effect of group, as comparing ADE vs. HC individuals was of primary interest, and in some cases also the effect of hemisphere and epoch. However, we did not perform exhaustive comparisons for all measures, in the interest of keeping the focus of our multi-level multi-task analysis on the hypothesis-driven questions specifically related to convergence of top-down and bottom-up processing.

Regarding the comment asking if we could compare the lateralization effect directly across task conditions (*i.e.,* is there a greater difference between hemispheres in the ISO task compared to VIA?): unfortunately, directly comparing signal intensity across tasks is not possible because the isoproterenol infusion induces physiological changes that can cause some dose-related signal reduction (we have attempted to address this in the past, e.g., Hassanpour et al, 2018 HumBrMapp). Consequently, our conclusions about spatial localization of top-down and bottom-up convergence are limited to group-level comparisons based on binary activation.

(2) A second 'major' relates to the intensity ratings (l. 530 ff). I found it very interesting that the ADE group reported higher cardiac, but lower exteroceptive intensity ratings during the VIA task. I understand the authors' approach to collapse within the ADE group, but it would be great to know which subgroup of patients drives this differential effect. It could be the case that the cardiac effect is predominantly present in the anxiety group, while the lower exteroceptive ratings are driven by the depression patients. Even if that were not the case, it would be highly instructive to understand the rating pattern within the anxiety group in greater detail. Do these patients 'just' selectively upregulate interoception, or is there even a perceived downregulation of exteroceptive signalling?

We have depicted these data below for reviewers’ reference, showing individual responses for each group (HC and ADE; panel A), as well as the ADE individuals separated by primary diagnosis (GAD = generalized anxiety disorder, n=24; AN = anorexia nervosa, n=16; MDD = major depressive disorder, n=6; panel B). When tested via linear regression, we found no differences in ratings across ADE subgroups (rating ~ subgroup * condition, F3=1.71, p=0.16 for main effect of subgroup). However, several factors should be considered in interpreting this result: first, all subgroups are small, particularly the MDD sample. Second, while these diagnostic labels refer to the most prominent symptom expression of each patient, every clinical participant in the study had a co-morbid disorder. Therefore, it is not possible to isolate disorder-specific pathology from our multi-diagnostic sample, and for this reason we refrained from including the subgroup-specific data in the manuscript.

**Author response image 2. sa4fig2:** (**A**) Post-trial ratings during the Visceral Interoceptive attention task, for reference. This is also shown in Figure 7D. (**B**) The same post-trial ratings in (A), but with the ADE group separated by primary diagnoses. Importantly, although assigned to one diagnostic category on the basis of most prominent symptom expression, most patients had one or more comorbidities across disorders. GAD = Generalized Anxiety Disorder. MDD = major depressive disorder. AN = anorexia nervosa. HC = healthy comparison.

l. 86: 'Conscious experience' of what, precisely? During the first round of reading, I was wondering about the extent to which consciousness as a general concept will play a role, which could be misleading.

We have changed it to “conscious experience of the inner body” in the text. The current study is limited in scope to the neurobiology of conscious perceptions of the inner body, not consciousness as a general phenomenon. We hope this distinction is now clear.

l.115: Particularly given the focus on predictive processing, I was wondering whether the (slightly outdated) spotlight metaphor is really needed here.

While not perfect, we believe it is still valid to metaphorically reference goal-directed attention towards the body as an “attentional spotlight”. Given the concern, we have minimized the focus on this metaphor, and the sentence now reads as follows:

“Extending beyond these model-based influences are goal-directed activities (also described previously as the ‘attentional spotlight’ effect [Brefczynski and DeYoe 1999]), whereby focusing voluntary attention towards certain environmental signals not only alters their conscious experience but selectively enhances neural activity in the responsive area of cortex.”

l. 129 ff: The sentence has three instances of 'and' in it, most likely a typo.

We have fixed this in the text.

l. 245: What do these ratings correspond to, i.e. what was the precise question/instruction?

The instructions for subjective ratings in each task are mentioned in the Methods (line 223 for ISO task, line 249 for the VIA task), and we have added more detail regarding the scale used to collect subjective intensity ratings.

l. 322: Could you provide the equation of the LMEM in the main text? It would be interesting to know e.g. whether participants/patients were included as a random effect.

We have provided this equation in the Methods (line 326).

l. 418 ff: I was confused about the statistical approach here. Why use separate t-tests instead of e.g. another LMEM which would adequately model task and condition factors?

We did not use t-tests, but instead used linear regression to look at differences in agranular PSC across groups, hemispheres, and epochs, as well as potential associations between PSC and trait measures. We have adjusted the wording in this Methods paragraph (line 418) to help clarity.

l. 425: As a general comment, it would be great to provide the underlying scripts openly through GitHub, OSF, ...

We agree with this comment, and our main analysis scripts have been posted on our OSF as an addition to the original preregistration of this work (https://osf.io/6nxa3/).

l. 443: For consistency, please report the degrees of freedom for the X² test.l. 454: ... and the F statistic would require two degrees of freedom (only the second is reported).l. 523: The t value is reported without degrees of freedom here (but has them in other instances).l. 540: Typo ('were showed').

We have reported degrees of freedom for all statistics.